**Precipitation stable isotopic signatures of tropical cyclones in Metropolitan**
**Manila, Philippines show significant negative isotopic excursions**
Dominik Jackisch[1], Bi Xuan Yeo[2], Adam D. Switzer[1,2], Shaoneng He[1], Danica Linda M.
Cantarero[3], Fernando P. Siringan[3] and Nathalie F. Goodkin[1,2,4]
[1] Earth Observatory of Singapore, Nanyang Technological University, Singapore

8  639798

[2] Asian School of the Environment, Nanyang Technological University, Singapore

10  639798

[3] Marine Science Institute, University of the Philippines Diliman, Quezon City 1101,
Philippines
[4] American Museum of Natural History, New York 10024, USA
Correspondence to: Adam D. Switzer (aswitzer@ntu.edu.sg)
**Abstract**
Tropical cyclones have devastating impacts on the environment, economies, and societies,
and may intensify in the coming decades due to climate change. Stable water isotopes serve
as tracers of the hydrological cycle, as isotope fractionation processes leave distinct
precipitation isotopic signatures. Here we present a record of daily precipitation isotope
measurements from March 2014 to October 2015 for Metropolitan Manila, a first of a kind
dataset for the Philippines and Southeast Asia. We show that precipitation isotopic variation
at our study site is closely related to tropical cyclones. The most negative shift in $\delta^{18}O$ value
(-13.84 ‰) leading to a clear isotopic signal was caused by Typhoon Rammasun, which
directly hit Metropolitan Manila. The average $\delta^{18}O$ value of precipitation associated with
tropical cyclones is -10.24 ‰, whereas the mean isotopic value for rainfall associated with
non-cyclone events is -5.29 ‰. Further, the closer the storm track to the sampling site, the
more negative the isotopic values, indicating that in-situ isotope measurements can provide
a direct linkage between isotopes and typhoon activities in the Philippines.

## 1. Introduction

The Philippine archipelago, with its fast-growing population clustered along the coastline, is one of the most vulnerable countries to climate change (Cinco et al., 2014). It is especially prone to the devastating effects of tropical cyclones. Thus, it is considered a hotspot region for hydrometeorological disasters (Cinco et al., 2014; Cruz et al., 2013; Takagi and Esteban, 2016). There is a clear need for developing a better understanding of tropical cyclone (TC) dynamics and cyclone histories in the context of prediction that may allow government agencies to implement proper mitigation and adaptation policies. Nine TCs per year made landfall on average between 1951 to 2013 in the Philippines. The number of TCs not making landfall but reaching Philippine waters is substantially higher with 19.4 per year (Cinco et al., 2016). Changing climate and associated warming of the surface ocean, will likely increase the intensity of tropical cyclones in the future (Emanuel, 2005; Webster and Holland, 2005; Woodruff et al., 2013).

The Philippines were struck by several devastating TCs in recent years (Table 1). Typhoon Haiyan (2013), which tracked over the Visayas has been the costliest TC to date (~ 2.06 billion USD in 2013), with strong winds and intense storm surges inundating coastal areas resulting, in more than 6000 fatalities (Alojado and Padua, 2015; Lagmay et al., 2015; Soria et al., 2016). Typhoon Rammasun, which made landfall in July 2014, is ranked number 3 with ~ 880 million USD in 2014 (Alojado and Padua, 2015; NDRRMC, 2014). Eighty percent of the strongest typhoons making landfall in the Philippines over the last three decades developed during higher than average sea surface temperatures (SST), which supports the hypothesis that TC intensities are projected to rise in the future with an increase in global temperatures (Guan et al., 2018; Webster and Holland, 2005; Takagi and Esteban, 2016). For example, SST was found to be anomalously high and reaching 29.6 °C during the formation of Typhoon Haiyan (Takagi and Esteban, 2016). The average Philippines' ocean SST for the period from 1945 to 2014 (basin between 6° − 18° N, 120° − 140° E) is ~ 28.5 °C based on National Oceanic and Atmospheric Administration Extended Reconstructed Sea Surface Temperature Dataset, Version 5 (NOAA ERSST v5) (Takagi and Esteban, 2016). By the end of the 21$^{st}$ century, average typhoon intensity in the low-latitude northwestern Pacific is predicted to increase by 14 % due to rising ocean temperatures (Mei et al., 2015).


A few studies have demonstrated the potential to investigate tropical cyclones using stable
water isotopes (Good et al., 2014; Lawrence et al., 2002; Munksgaard et al., 2015; Pape et al.,
2010). As dynamic tracers of hydrological processes, stable water isotopes ($\delta^2$H and $\delta^{18}$O) can
provide insights into the water and energy budgets of TCs (Good et al., 2014; Lawrence and
Gedzelman, 1996). In the regions with general TC occurrence, significantly lower $\delta^2$H and $\delta^{18}$O
are associated with TC rainfall due to strong isotope fractionation processes, compared to
other tropical rain events (Lawrence, 1998; Lawrence and Gedzelman, 1996). Furthermore,
$\delta^2$H and $\delta^{18}$O have been used successfully to interpret TC history from paleoarchives, such as
tree rings and speleothems (Oliva et al., 2017). For instance, tree-ring cellulose isotope
proxies have recorded the recent 220 years of cyclones in the southeastern USA (Miller et al.,
2006); similarly, high-resolution isotopic analysis of tree-rings from the eastern US revealed
the occurrence of hurricanes in 2004 (Li et al., 2011); a 23-year stalagmite record from Central
America was used to reconstruct past TC activity (Frappier et al., 2007), and isotope signals
from a 800-year stalagmite record were used to reconstruct past TC frequencies in
northeastern Australia (Nott et al., 2007). Interpretation of TC history in paleotempestology
from paleoarchives is based on the fact that TCs leave distinct isotopic signatures on
precipitation, possibly providing information on TC's evolution and structure (Lawrence et al.,

2002).


The depletion in stable isotopes has been attributed to the high condensation levels, strong
isotopic exchanges between inflowing water vapour and falling raindrops in cyclonic rainfall
bands, resulting in a temporal decrease of isotopic values throughout a rain event (i.e.
amount effect) (Lawrence, 1998; Lawrence and Gedzelman, 1996). Isotopic depletion can be
further enhanced by TC's thick, deep clouds, relatively large storm size and longevity
(Lawrence, 1998). Furthermore, while isotopic depletion increases inwards towards the eye
wall of the storm (Lawrence and Gedzelman, 1996), isotope ratios inside the inner eye wall
region are relatively enriched, likely due to an intensive isotopic moisture recharge with heavy
isotopes from sea spray (Fudeyasu et al., 2008; Gedzelman et al., 2003). These findings are
based on work conducted in the 1990s in Puerto Rico and on the southern and eastern coasts
of the United States. More recently, these previous findings have been confirmed by studying
TCs which occurred in a few other regions, such as in China or Australia (Chakraborty et al.,
2016; Fudeyasu et al., 2008; Good et al., 2014; Munksgaard et al., 2015; Xu et al., 2019).

The above-mentioned studies are geographically limited to a few locations globally, with no
studies in Southeast Asia and the Philippines in particular. Here, we present the first such
study for the Philippines, with daily isotope measurements of precipitation from
Metropolitan Manila (the National Capital Region) spanning from March 2014 to October
2015. During the study period, nine tropical cyclones passed by or made landfall within 500
km of the sampling site (Fig. 1). The main objectives of this research are the following:
-   To understand if there is an isotopic variation in precipitation associated to the TC

landfall in the Philippines and if tropical cyclones leave clear isotopic signals.

-   To identify the isotopic signals measured for Metropolitan Manila and the intensity of

the isotopic depletion associated to TC activities, and to identify how it is represented

spatially.

-   To understand the isotopic variation with distance from the TC track in the Philippines.
Our findings provide a baseline dataset for reconstruction of typhoon activities using stable
isotopes and contribute to a better understanding of past and future TC activities in the
Philippines.


**2.   Materials and methods**

**2.1 Site description**

The Philippines is a Southeast Asian country comprising more than 7000 islands located in the
Northwest Pacific between 4° 40' N and 21° 10' N, and 116° 40' E and 126° 34' E (Fig. 1). The
country experiences an average annual rainfall of about 2000 mm, influenced by two
monsoon seasons, the northeast monsoon from November to April and the southwest
monsoon from May to October (Cinco et al., 2014). About 35 % of the annual rainfall is related
to TC activity, while its contribution rises to about 50 % for Luzon and decreases to 4 % for
the southern island of Mindanao (Cinco et al., 2016). Part of the rainfall amount in the
Philippines is of orographic nature due to north-south oriented mountain ranges of more than
1000 m spanning the largest islands of Luzon and Mindanao (Villafuerte et al., 2014). The
majority of the steadily growing population in the Philippines (101 million 2017 census) live
in densely populated, low-elevation areas close to the coastlines (Cinco et al., 2014, 2016;
Philippine Statistics Authority, 2017).


**2.2 Isotopic data**

In total, 186 daily precipitation samples were collected from 10 March 2014 to 26 October
2015 using a PALMEX collector (Gröning et al., 2012) at the Marine Science Institute of the
University of the Philippines Diliman located in Quezon City, which is a part of Metropolitan
Manila. The rain station was installed on the rooftop of the Marine Science Institute
(14°39'02.5"N, 121°04'08.6"E), which is centrally situated in the campus and surrounded by
trees and various green spaces. The rooftop location proofed ideal for rainwater collection as
it allowed for unobstructed access to rainwater without any potential sources of
contamination. Samples were collected daily at 10 am, and transferred without headspace to
30-ml HDPE bottles for storage prior to analysis. Samples were sent to the Earth Observatory
of Singapore, Nanyang Technological University, Singapore and were analyzed for stable
isotopes using a Picarro L1230-*i* laser spectroscopy instrument. We followed the procedures
described by Van Geldern and Barth (2012) for post-run corrections and calibration. Three in-
house water standards used for calibration include KONA (0.02 ‰ of $\delta^{18}$O; 0.25 ‰ of $\delta^2$H),
TIBET (-19.11 ‰ of $\delta^{18}$O; -143.60 ‰ of $\delta^2$H), and ELGA (-4.25 ‰ of $\delta^{18}$O; -27.16 ‰ of $\delta^2$H).
They are calibrated against the international reference water VSMOW2 and SLAP2. Long-term
analysis of our QA/QC standards yields precision of 0.04 ‰ for $\delta^{18}$O and 0.2 ‰ for $\delta^2$H.  We
used $\delta^{18}$O and $\delta^2$H to calculate deuterium excess, which is defined as d-excess = $\delta^2$H $-$ 8$*$ $\delta^{18}$O
and is commonly regarded to reflect evaporation conditions of moisture source regions.

**2.3 Cyclone track data**

The International Best Track Archive for Climate Stewardship (IBTrACS) dataset contains
global TC best-track data, and is a joint effort of various regional meteorological institutions
and centers that are part of the World Meteorological Organization (WMO). The data is
publicly available, and comprises information on storm eye/center with its coordinates, wind
speed, and pressure, etc., with a temporal resolution of six hours (Knapp et al., 2010; Rios
Gaona et al., 2018). Apart from visualization of cyclone paths, we used the dataset to calculate
the spatial distance between the storm's eye coordinates and our sampling site.

**2.4 Satellite precipitation data**

We used the IMERG Version 5 Final daily product, a remotely-sensed precipitation dataset
from satellites to highlight cyclonic tracks and precipitation patterns of several TC's passing
by Metropolitan Manila, and to identify which rainfall events were not affected by cyclonic
activity, and instead were associated with local or other regional convection activities. Such
dataset is beneficial as it provides quasi-global grid-based rainfall estimates for land and the
oceans (Poméon et al., 2017). The Integrated Multi Satellite Retrievals for GPM (IMERG) from
the Global Precipitation Measurement (GPM) programme with a fine 0.1-degree grid size
(Huffman et al., 2017) has been available since March 2014, and provides precipitation data
in different temporal resolutions, such as half-hourly or daily. Such satellite rainfall data has
been previously utilized to show TC tracks and related rainfall intensities (Rios Gaona et al.,
2018; Villarini et al., 2011).

**2.5 Rainfall, temperature and relative humidity data**

Daily rainfall, mean daily relative humidity and mean daily temperature data was obtained
from the Philippine Atmospheric, Geophysical and Astronomical Services Administration
(PAGASA), which maintains a rainfall monitoring station about 2.7 km away from our sampling
site. The data is freely available for the period 2013 to 2017, and can be accessed on the
Philippines Freedom of Information website (www.foi.gov.ph).


**3. Results**
**3.1 Isotopic variation of stable isotopes in daily precipitation**

One hundred and eighty-six daily precipitation samples were collected during the 19 months
of the study period spanning from 10 March 2014 to 27 October 2015 in Metropolitan Manila.
Their stable isotope compositions show large seasonal isotopic variability; $\delta^{18}O$ ranges from
4 ‰ to -13.84 ‰, and $\delta^2H$ from 16.84 ‰ to -99.1 ‰ (Fig. 2). The highest $\delta^{18}O$ of 4 ‰ was
observed on 9 April 2014 during the annual dry period, whereas the lowest $\delta^{18}O$ of -13.84 ‰
was observed on 16 September 2014 in association with TC activity. The mean $\delta^{18}O$ of
precipitation at the study site is -5.29 ‰ for non-TC rain systems, while TCs, as large regional
convective systems, have the potential to cause a change in δ-values of up to almost 9 ‰
relative to the mean. The average $\delta^{18}O$ of the nine TCs that tracked within <500 km from the
sampling site is -10.24 ‰ (STDEV of 2.11), a factor of 2 larger than the mean from non-TC
precipitation (average is -5.29 ‰, STDEV of 2.64).

An inter-annual variation of stable isotopes in precipitation is observed in the time series of
Metropolitan Manila, where the generally humid summer months are characterized by heavy
rainfall and exhibit lower isotope values compared to the rest of the year (Fig. 2). The
precipitation isotopes are chararacterized by slightly higher values during winter and spring,
when temperatures and relative humidity are lower with less frequent rainfall. Especially
early 2015 shows drier conditions with sporadic rainfall and relative humidity levels of about
60 % to 70 %. This is also reflected in the precipitation collected on 1 March 2015 with $\delta^{18}O$
of 0.01 ‰ and $\delta^2H$ of 9.8 ‰, respectively. Although d-excess shows relatively high temporal
variability, ranging from -15.18 ‰ to 24.31 ‰, it largely clusters in a small range between 5
‰ to 15 ‰.

Based on the daily isotope measurements of rainfall events between 2014 and 2015, we
determined the LMWL (local meteoric water line) for the study site as $\delta^2H= 7.2674 \times \delta^{18}O +$
5.4103 (Fig. 3), indicating that slope and intercept of the LMWL are lower due to the influence
of tropical precipitation compared to the GMWL (global meteoric water line) with $\delta^2H= 8 \times$
$\delta^{18}O + 10$ (Craig, 1961).

In order to assess meteorological controls on the isotopic composition of daily precipitation
at Metropolitan Manila, we investigated the correlation between $\delta^{18}O$, daily precipitation
amount, daily mean temperature, and daily mean relative humidity. Additionally, $\delta^{18}O$ is
compared to d-excess (n=187) (Fig. 4). We found that $\delta^{18}O$ is weakly correlated to d-excess
($R^2$=0.2187), precipitation amount ($R^2$=0.1087), and relative humidity ($R^2$=0.1323). No
association is observed between $\delta^{18}O$ and temperature ($R^2$=0.0338).

In order to get further insights into the seasonal variations, we also calculated the average
values for each month in the time series for every isotopic and climatic parameter, while
rainfall is reported as monthly totals (Table 2). $\delta^{18}O$ is relatively low during the summer
months, for instance with -7.29 ‰ in September 2014 compared to the months of winter and
spring with -0.53 ‰ in April 2014 or -0.66 ‰ in February 2015. Similarly, the monthly rainfall
total is less in winter and spring with 19.2 mm in March 2014 and 29.2 mm in January 2015
compared to the summer months such as July and August 2014 with 455.4 mm and 420.7 mm
respectively. As mentioned before regarding the daily measurements, we also observe on the
monthly scale conditions which are more humid in the summer. We investigated the
relationship between the isotopic composition of precipitation ($\delta^{18}O$) and meteorological
parameters (total monthly rainfall, average relative humidity and temperature) on a monthly
scale. $\delta^{18}O$ and $\delta^2H$ are strongly correlated (Pearson correlation coefficient) with r=0.96
(n=18, p-value=<0.0001 and 99% confidence level), whereas the relationship between $\delta^{18}O$
and d-excess yields an r of -0.64 (n=18, p-value=0.003). A clear negative correlation was
determined between $\delta^{18}O$ and precipitation with r=-0.67 (n=18, p-value=0.002) and between
$\delta^{18}O$ and relative humidity with r=-0.85 (n=18, p-value=<0.0001). $\delta^{18}O$ and temperature are
not correlated with r=0.04 (n=18, p-value=0.87).

A relationship between isotopic value and the distance of the TC towards the sampling site
was found. The TCs' distance of up to 500 km to sampling site and the precipitation isotope
value are correlated with r=0.55 (n=16, p-value=<0.05 and 99% confidence level). This
relationship weakens with an increase in the distance from the sampling site: a distance of
500 to 1000 km yields an r of 0.2 (n=19, p-value=0.41), the distance of 1000 to 1500 km yields
an r of 0.18 (n=24, p-value=0.40), while a 1500 to 2000 km distance results in an r of 0.1 (n=21,
p-value=0.69).

**3.2 Precipitation isotope evolution during TC events**

Overall, precipitation isotopes associated with TCs mark the lower range of $\delta^{18}O$ values during
the study period. Especially during the 2014 season, precipitation with low isotope values
mostly occurred throughout passage of TCs. For instance, Rammasun led to the lowest δ-
value (Fig. 5, point a, -13.84 ‰) of the whole study period, while other TCs such as Fung-
Wong (Fig. 5, point c, -12.16 ‰), Kalmaegi (Fig. 5, point b, -11.39 ‰), or Hagupit (Fig. 5, point
d, -9.88 ‰) caused other negative excursions in isotopic values. The 2015 season is
characterized by on average a slightly higher isotopic enrichment during the summer months
with heavy rainfall. Nonetheless, a similar noticeable isotope signal is visible with low $\delta^{18}O$,
clustered along the lower end of the sample range, for example, caused by Linfa (Fig. 5, point
f, -8.5 ‰) or Koppu (Fig. 5, point i, -8.7 ‰). The other TCs that occurred during the study
period and were investigated by us were Mekkhala (Fig. 5, point e, -10.77 ‰), Twelve (Fig. 5,
point g, -7.7 ‰) and Mujigae (Fig. 5, point h, -7.5 ‰). However, relatively negative isotope
samples (Fig. 5) also originated from non-TC rainfall systems. Those events are discussed
below.

Out of the nine TCs that occurred within a 500 km radius from the sampling site, Rammasun
and Kalmaegi left clearly observable, distinct isotopic signatures during their approach and
dissipation, which we will therefore present in more detail in the next paragraphs. Typhoon
Hagupit (Fig. 5, point d) similarly led to a clear isotopic evolution pattern during its time of
occurrence in the Philippines and is shown in the supplementary (S1).

Typhoon Rammasun's rainfall intensities based on the IMERG precipitation data together
with its track from IBTrACKS is shown in Fig. 6a. Typhoon Rammasun stands out in our study
period as it moved straight towards the National Capital Region of the Philippines, resulting
in a direct hit. Rammasun, locally named Glenda, made landfall in the Bicol region of southern
Luzon on 15 July, with wind speeds of about 160 km/h. On 16 July, it passed south of
Metropolitan Manila 50 km from our sampling site, with maximum winds of 130 km/h,
gradually losing strength over land. As Rammasun approached on 15 July, the precipitation
exhibited relatively high $\delta^{18}O$ of -4 ‰ while rainfall was weak (Fig. 7a). On 16 July, $\delta^{18}O$ shifted
to -13.84 ‰, while the typhoon's track was the closest to our sampling site and rainfall
amount was high. As Rammasun moved away, precipitation isotopes became more positive,
and rainfall amount decreased. The characteristic isotopic evolution with time related to
Rammasun's distance and rainfall intensities can be seen in Fig. 8a, where the different radii
indicate the distance to the sampling site, and the strong isotopic depletion observed on 16
July is also evident. As Rammasun with its storm center tracked towards the northwest and
away from Metropolitan Manila, our precipitation samples were relatively isotopically
enriched for the following two days, namely -9,12 ‰ on 17 July and -6,26 ‰ on 18 July.

Typhoon Kalmaegi, locally named Luis, was the first typhoon to make landfall in the
Philippines two months after Rammasun. Kalmaegi reached typhoon intensity on 13
September, making landfall the following day in northern Luzon, with maximum wind speeds
of about 120 km/h. Kalmaegi tracked relatively far away from the sampling site (about 350
km), but the accumulated rainfall it produced was centered south of the track, placing it
considerably closer to the National Capital Region (Fig. 6b). Despite the distance of the eye
from the sampling site, a characteristic isotopic pattern was visible, with the most negative
$\delta^{18}O$ value of -11.39 ‰ on 15 September, coincident with the highest rainfall (Fig. 7b). The
following day, $\delta^{18}O$ values returned to higher values with the increase in distance from the
eye. This is also seen in a spatial representation in Fig. 8b, visualizing the track of Kalmaegi
and the respective $\delta^{18}O$ values. Kalmaegi was first approaching the sampling site on 14
September and passed away on 15 and 16 September. The lowest $\delta^{18}O$ was observed on 15
September and is indicated in the figure in dark blue.


**4.   Discussion**
**4.1 Stable isotopes of precipitation – a possible tracer for TCs**

As stable water isotopes fractionate during the physical process of evaporation and
condensation, they serve as effective tracers in the hydrological cycle (Dansgaard, 1964; He
et al., 2018; Risi et al., 2008; Tremoy et al., 2014). Here, we have demonstrated that stable
water isotopes can possibly be used to identify TC activity in the Southeast Asian region by
excursions in $\delta^{18}O$, providing evidence and supporting the hypothesis that TCs may leave a
clear isotopic signal in the Philippines. The strong isotopic depletion is due to high
condensation efficiencies in cyclonic convective rain bands, leading to extensive
fractionation. This is particularly pronounced in intense, large-scale TCs (Lawrence, 1998;
Lawrence and Gedzelman, 1996). In the previous section, we have presented our findings of
precipitation isotope ratios associated with typhoon activities affecting Metropolitan Manila
during the study period of March 2014 to October 2015. Based on our time series, we
therefore argue that for the Philippines, the lowest measured isotope value likely indicates
the occurrence of a TC, such as is the case for Typhoon Rammasun (Fig. 5). Similarly, other
anomalously low $\delta^{18}O$ values at our site are caused by TC making landfall or passing by.

Individual TCs (Rammasun and Kalmaegi) were characterized by consistent isotopic
excursions to very negative $\delta^{18}O$ in a range of up to -9 ‰ compared to the mean isotopic
value of -5.29 ‰ (Fig. 7 and 8). A TC approaching the sampling site had relatively higher
isotope values than at its later stages when it was closest to the site in Metropolitan Manila.
When at its closest, strong rainfall together with increased fractionation depleted
precipitation isotopes, leading to a distinct drop in isotope value. Such a strong negative
isotopic shift in precipitation has been previously observed in other regions (Fudeyasu et al.,
2008; Lawrence and Gedzelman, 1996; Munksgaard et al., 2015; Xu et al., 2019). As the TC
moved away and rainfall intensities weakened, $\delta^{18}O$ in precipitation became again more
positive, likely due to evaporative effects (Munksgaard et al., 2015; Xu et al., 2019).

As the strongest TC in terms of wind speeds, damage costs, and fatalities, Typhoon Rammasun
reduced $\delta^{18}O$ most during our study period, to -13.84 ‰. Similarly, Typhoon Kalmaegi led to
extensive damage and caused a significantly negative excursion in precipitation $\delta^{18}O$ to -11.39
‰, suggesting that the lowest isotope values might indicate the occurrence of the strongest
TC at that time at our site in the Philippines. We note that our isotopic measurements are
similar to observations elsewhere. For example, the range of $\delta^{18}O$ values caused by Typhoon
Shanshan affecting the subtropical Ishigaki island was -6 to -13 ‰, (Fudeyasu et al., 2008);
Tropical Cyclone Ita led to a range of -4.8 to -20.2 ‰ in northeastern Australia (Munksgaard
et al., 2015); several TCs which made landfall in Texas resulted in isotope values from -3.9 to
-14.3 ‰ (Lawrence and Gedzelman, 1996); or hurricanes that affected Puerto Rico and
southern Texas were found to deplete $\delta^{18}O$ up to -18 ‰ (Lawrence, 1998). The lowest value
resulting from Typhoon Phailin on the Andaman Islands was reported to be -5.5 ‰, and
Typhoon Lehar depleted the precipitation sample to -17.1 ‰ (Chakraborty et al., 2016). For
TCs within a distance of up to 500 km from the sampling site at the University of the
Philippines Diliman in Metropolitan Manila we measured an isotopic range of -7.7 ‰
(Typhoon Koppu) to -13.84 ‰ (Typhoon Rammasun). Despite the overall comparability to our
measurements, differences exist. The lowest values observed in some studies are
considerably more negative than at our site (Lawrence, 1998; Munksgaard et al., 2015).
However, we attribute these differences to a variety of features, such as the specific climatic
condition at each site, differences in temperature, humidity, and altitude or latitude, which
are likely contributing factors to the observed isotopic variation by altering isotopic
fractionation. Further, rainout history, location of typhoon tracks, topography, respective
strength of each TC, as well as its distance to the sampling site most likely have a significant
influence as well (Fudeyasu et al., 2008; Good et al., 2014; Munksgaard et al., 2015; Xu et al.,

360 2019).


We used IMERG satellite precipitation data to assess why other very low isotopic excursions
occurred on various days (Fig. 5). IMERG data with its fine spatio-temporal resolution allows
the identification of convective rainfall areas and the passage of TCs and other rain systems
(Fig. 6). Our analysis shows that precipitation events with anomalously low isotope signals
unassociated with TCs are largely related to local, strong convective rainfall events or large
scale and slow-moving rain areas passing over the National Capital Region. Therefore, the
degree of convection is responsible for the other observed low $\delta^{18}O$ outliers that are not
related to cyclone rainfall, as strong convection and long stratiform rainfall leads to intense
fractionation (He et al., 2018; Risi et al., 2008; Tremoy et al., 2014). Contrarily, we speculate
that the more positive isotope values clustering along the higher end of the sample spectrum
around 0 ‰, are associated with local, short convective rainfall events and light intensity rain
as confirmed with IMERG satellite precipitation data. Additionally, the PAGASA rain gauge
data indicates that rainfall amounts are very low during days with such very enriched isotope
samples, such as 0.3 mm/day for the highest recorded sample of 4 ‰ on 9 April 2014.
Interestingly, TCs at our site were found to be related with low isotope values together with
high rainfall amounts (Fig. 5), while the majority of other low isotopic values unassociated
with TCs were characterized by on average lesser rainfall amounts. This possibly indicates that
TCs in the Philippines, besides using for instance modern-day satellite or radar data, can be
detected using these two parameters, i.e. strong isotopic depletion coupled with high rainfall
amounts.

The aforementioned local convective precipitation events have the potential to induce a signal of very negative $\delta^{18}O$, which is not related to TC activities. We therefore label  such a signal as a "false non-TC signal", as it is induced by non-TC rainfall. This results in the fact that TCs occurring during our study period do not entirely cluster along the lowest range of isotope values as seen in figure Fig. 5. Nevertheless, Typhoon Rammasun caused a clear drop in $\delta^{18}O$ and stands out in the dataset. This might be the case because Rammasun's track and heavy rainfall comes in closest proximity (50 km) to the sampling site. Other TCs occurring within the 500 km radius did not lead to such a clear negative isotopic signature, likely because these typhoons did not pass the sampling site at all or heavy rainfall occurred elsewhere within the TC rainfall system (see S 2 for their tracks and accumulated rainfall areas). Some of these TCs have intense rainfall areas over other parts of the Philippines and are characterized by a variable track, likely influenced by land interactions. Land interaction reduces TC strength and can lead to rain out due to orographic effects induced by the north-south oriented mountain ranges (Park et al., 2017; Xie and Zhang, 2012; Xu et al., 2019). Especially Typhoon Koppu rained out before making landfall and abruptly changed its track, instead of passing by the Metropolitan Manila. Similarly, Typhoon Mekkhala's intense rainfall occurred along the eastern coasts, before it started to dissipate. Evidently, due to these factors the isotope values associated with those TCs were not as negative as during Rammasun. Therefore, a TC, which is relatively far away from the sampling site, produces an isotope signal that is not as clear and as negative, thus averaging out between the other low values from rain systems unassociated with TC.

**4.2 Drivers of isotopic variation at Metropolitan Manila**

$\delta^{18}O$, $\delta^2H$, and the second parameter d-excess all show seasonal variabilities and are influenced by several climatic factors, including precipitation amount, temperature and relative humidity. The scale of their influence varies depending on daily or monthly values. The results indicate that $\delta^{18}O$ on daily levels is not influenced by temperature, relative humidity or precipitation amount (Fig. 2) as drivers of isotopic variability. Instead, we speculate that other processes, such as large scale convection and processes at the moisture source region might influence stable isotopes of precipitation at our study site (Conroy et al.,

2016; He et al., 2018; Kurita, 2013). Interestingly, $\delta^{18}O$ is not affected by precipitation amount
on short timescales (Fig. 4), which has also been previously confirmed in other tropical
regions, suggesting that the tropial amount effect is not reflected on daily timescales
(Belgaman et al., 2016; Dansgaard, 1964; He et al., 2018; Kurita et al., 2009; Marryanna et al.,
2017; Permana et al., 2016). However, comparing monthly $\delta^{18}O$ to $\delta^2H$ and d-excess and to
monthly average precipitation, relative humidity and temperature, the results are clearly
different (Table 2). These monthly observations show close relationships with each other,
especially $\delta^{18}O$ and precipitation amount are linked (see section 3.1). The close relationship
between these two parameters can be attributed to the tropical amount effect (Aggarwal et
al., 2012; Bowen, 2008; Conroy et al., 2016). The relatively close relationship with r=-0.67
between monthly $\delta^{18}O$ and monthly total precipitation might be likely due to the influence of
regional convective activities on the isotopic composition of precipition (Bony et al., 2008; He
et al., 2018; Moerman et al., 2013; Risi et al., 2008).

**4.3 Distance of TCs from Metropolitan Manila**

Our observations provide details on spatial distance from the collection site towards TCs'
centers, as our findings indicate that the distance from the storm center to the sampling site
impacts the isotopic value (see section 3.1). This suggests that a TC more than 500 km away
from the sampling site has no influence on precipitation isotopes (Munksgaard et al., 2015).
Thus, the closer the TC is to the sampling site, the more negative the isotope signal and the
larger the δ-change. This relationship might provide information on storm structure and
intensity, as the intensity increases with proximity of the TC to the sampling location. We thus
confirm that the isotope value at our location is a function of the closest approach of the
storm's center to the sampling site (Lawrence and Gedzelman, 1996).

Figure 8 displays all the precipitation samples associated with TC presence and activities
within a 2000 km radius from Metropolitan Manila, and further highlights the relationship
between distance and isotopic depletion, additionally providing a spatial indication of TC's
quadrants and their tracks relative to the location of the sampling site. Strongest depletion
occurs within the 500 km radius. However, two relatively negative outliers are located within
a 1000 to 1500 km radius in the northwest quadrant (see points a and b in Fig. 9). These two
samples were taken during the passage of tropical storm Kujira on 22$^{nd}$ and 23$^{rd}$ of June 2015
(Fig. 5), which was more than 1000 km away from Metropolitan Manila travelling east along
the coast of Vietnam as seen with IBTrACKS data. We investigated these two samples with
IMERG satellite precipitation data and identified them as a part of a mesoscale system, with
strong convective cells delivering intense rainfall, leading to distinct isotopic depletion and
inducing a "false non-TC signal" of very negative $\delta^{18}$O, which is not related to TC activity.

**4.4 Cyclone track's rainfall intensity**

IMERG satellite precipitation data also reveal that the highest rainfall intensities occur at the
left side of the TC track for all the TC within the 500 km radius, except for Hagupit and
Mekkhala, which are more complex cases (Fig. 6a, b, supplementary S 2). This is in contrast
to the results from Villarini et al. (2011), who found that the largest rainfall accumulation
appeared on the right side of the hurricane tracks. They also noted that large rainfall amounts
occured far away from the storm's track, which we can confirm and quantify with our
observations. The largest rainfall totals vary in a range of 50 to 150 km away from the storm's
center depending on the TC. For Kalmaegi the intense rainfall areas are up to 150 km away
from the storm's center. These areas with the highest rainfall totals should most likely
coincide with the most negative isotope value, indicating that the strongest depletion occurs
in the outer cyclonic rain bands. This is consistent with previous findings (Gedzelman et al.,
2003; Lawrence and Gedzelman, 1996; Munksgaard et al., 2015). However, Fudeyasu et al.
(2008) observed the highest isotope values in the inner eye wall, i.e. in close proximity to the
storm's center. We could not investigate this further as no TC passed by our site in a distance
of about 20 km, which is the size of a typical typhoon's eye (Weatherford and Gray, 1988).

**4.5 Implications for paleoclimate studies**

Isotope proxies from paleoarchives such as tree rings and speleothems have been utilized to
reconstruct past cyclone activities (Frappier, 2013; Frappier et al., 2007; Miller et al., 2006;
Nott et al., 2007). For instance, stalagmites yielded a record of weekly temporal resolution
with negative isotopic excursions related to TC activity (Frappier et al., 2007). Such a high
temporal resolution from stalagmites makes our in-situ measurements very comparable,
highlighting the potential to use both in conjunction. Similarly, high-resolution tree ring
isotope analysis identified the occurrence of Hurricane Ivan and Hurricane Frances in 2004,
which both resulted in the lowest observed precipitation isotope values for that year (Li et
al., 2011). Nevertheless, it is important to consider possible limitations at the study site that
arise in paleotempestology, such as sea level change or disruption of sedimentological
records through floods or tsunamis. These need to be evaluated when comparing
precipitation isotopes related to TCs with other proxy records such as speleothems and
coastal deposits and when choosing the study area (Oliva et al., 2017). However, the
aforementioned paleotempestology studies suffer from uncertainty regarding parameters
such as TC intensity and distance to the storm's center affecting the isotope signal. Our study
provides further information on these parameters as we hypothesize that immediate
proximity of a TC results in very low $\delta^{18}O$. Therefore, we might aid with a better interpretation
of paleoarchives. Moreover, these studies are limited in number and only focus on a few
regions affected by TCs, such as Central America and the Southeastern USA (Frappier et al.,
2007; Miller et al., 2006). However, more paleotempestology studies investigating
paleoarchives related to typhoon footprints covering different regions and countries would
provide a better understanding of past TC activity, ultimately resulting in better and more
accurate climate reconstructions. TC projections related to climate change could also be
improved, which is especially relevant for decision makers dealing with TC related impacts
and damages. Our in-situ isotope measurements provide baseline data input in an
understudied tropical region, providing isotopic data of TC occurrence and quantifying the
isotopic depletion associated with TC activity. Further, our 19-month dataset suggests that
the lowest measured isotope value at the Philippines study site is associated with TC activity,
resulting in the distinct negative isotopic shift in the time series (Fig. 5). As rain out history,
topography, distance of track or rainfall unassociated with TCs can induce a weak or "false
non-TC signal", it is important to choose stalagmites or trees as archives based on their
location, ideally covering a spatial gradient thus capturing a TC in its full size.


**5. Conclusions**

Our study demonstrated that a strong, high-energy TC with a track directly approaching and
hitting the sampling site leads to a clear isotopic signal in a time series in the Philippines. If
the TC is further away, such as more than 500 km from the site, or heavy TC rainfall occurred
elsewhere prior of making landfall, the signal is not as clear and might average out between
other rainfall events. Other strong convective rainfall events unassociated with TCs may result
in similarly low isotope values, and we label these as a weak or "false non-TC signal".
Therefore, the distance of TC to the sampling site is a key factor in influencing the isotope
signal and that such a spatial component needs to be considered when interpreting the
isotope signal. However, a longer time series isotope record would help to better constrain
controlling factors, such as the influence of topography on high-energy TCs. To what extent
mountain ranges and low elevation coastal areas shape the TC induced isotope signal needs
further investigation. Based on our findings we conclude that the location of precipitation
sample collection needs to be chosen strategically. Ideally, several rainwater collection
stations should be operated, covering a wide geographical range such as stretching from
northern Luzon to its south. With such a spatial gradient coverage, a TC would likely be
captured in its full size. Consequently, we aim to expand our time series spatially and
temporally.

Our dataset is the first of such record in the Philippines and provides much needed data in
scarcely sampled Southeast Asia. It can be used as a baseline in paleotempestology studies
reconstructing past TC history, in conjunction with tree ring and speleothem datasets, as our
data suggest that for Metropolitan Manila the lowest measured isotope value is caused by
typhoon activity. A higher precipitation sampling frequency on sub-daily levels at several
locations would yield more detailed constraints on TC parameters such as storm structure,
which we aim to realize in the future.


**Data availability**

The underlying research data can be accessed via the supplementary document.


**Author Contributions**

Dominik Jackisch analyzed the data and wrote the manuscript. Bi Xuan Yeo contributed to
data analysis and improved the manuscript. Adam D. Switzer conceived the idea, reviewed
and improved the manuscript. Shaoneng He provided advice, reviewed and improved the
manuscript. Danica Cantarero and Fernando P. Siringan collected the precipitation samples
and improved the manuscript. Nathalie F. Goodkin reviewed and improved the manuscript.


**Competing interests**

The authors declare that they have no conflict of interest.


**Acknowledgments**

This study is supported by the National Research Foundation Singapore and the Singapore
Ministry of Education under the Research Centres of Excellence Initiative. It is Earth
Observatory of Singapore contribution no. 188. This study is also the part of IAEA Coordinated
Research Project (CRP Code: F31004) on "Stable isotopes in precipitation and palaeoclimatic
archives in tropical areas to improve regional hydrological and climatic impact model" with
IAEA Research Agreement No. 17980.

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

**Figures and Captions**

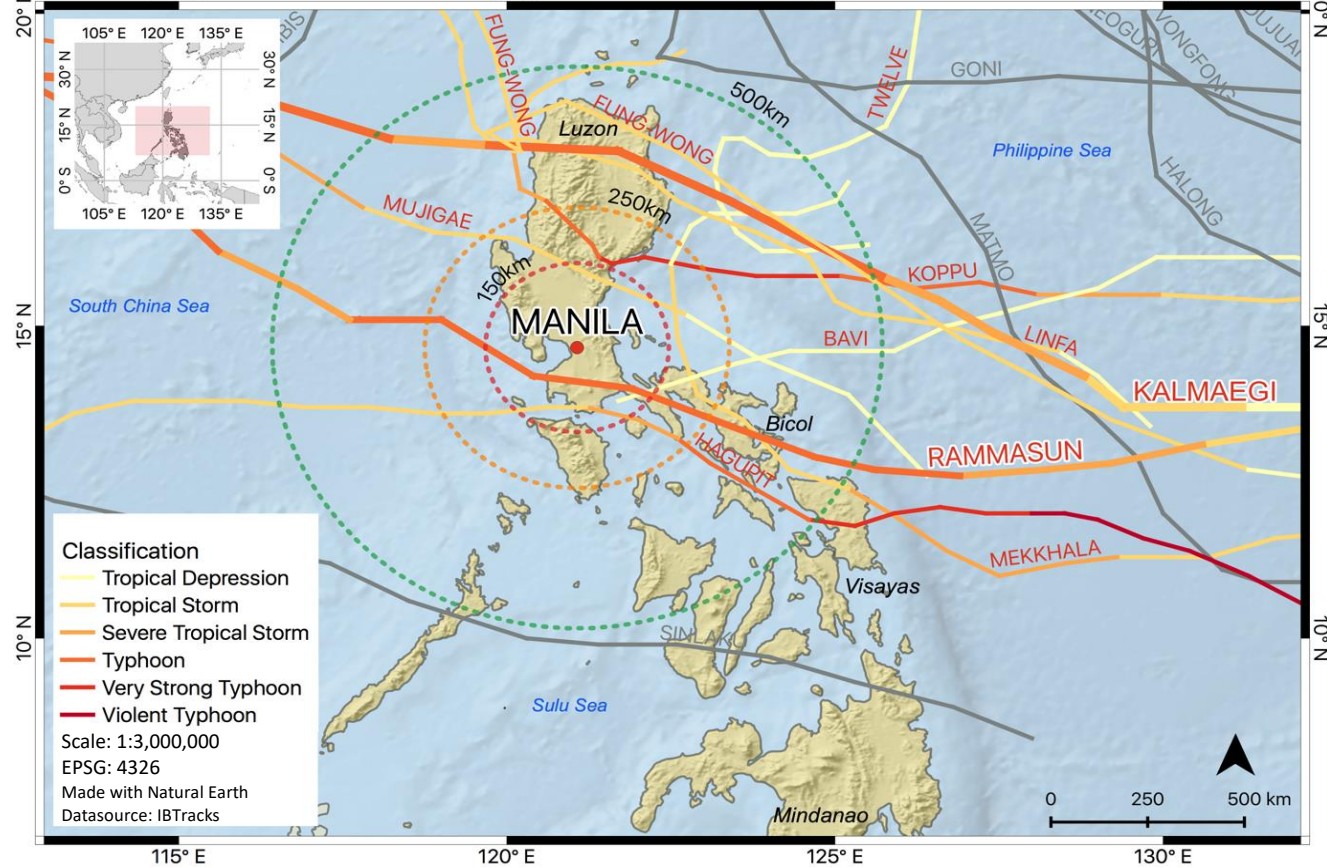

**Figure 1 Metropolitan Manila sampling site and TC tracks of 2014 and 2015 seasons.** Three different sized circles indicate the distance to the sampling site with the outermost one being 500 km in radius. Cyclone tracks are color coded according to the typhoon classification from Regional Specialized Meteorological Center (RSMC) Tokyo. Cyclones in gray color refer to TC outside the 500 km radius.

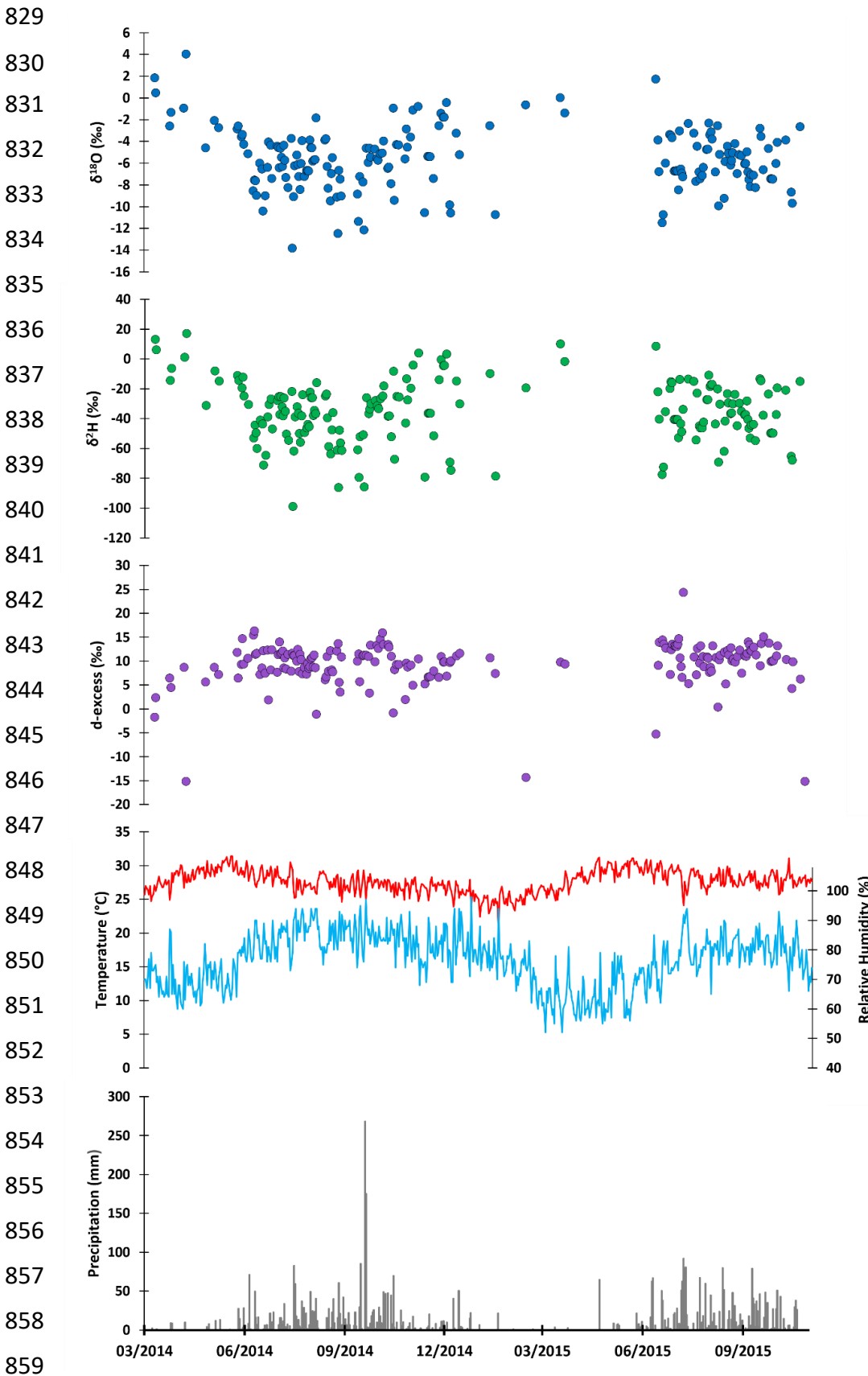

**Figure 2 Time series of daily variations of δ¹⁸O, δ²H, d-excess, temperature, relative humidity and precipitation amount** at Metropolitan Manila, Philippines.

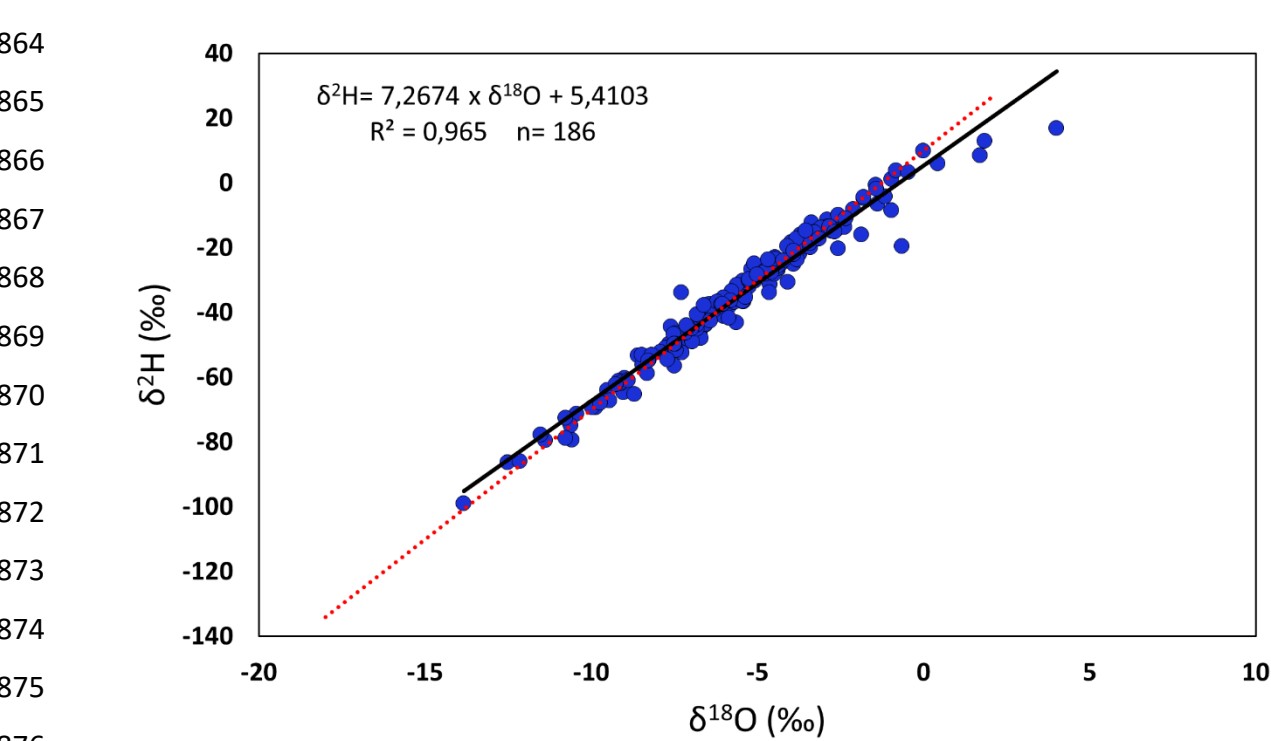

**Figure 3 Local meteoric water line (LMWL)** established for Metropolitan Manila, Philippines. The red dotted line represents the global meteoric water line (GMWL) ($\delta^2H= 8 \times \delta^{18}O + 10$; Craig, 1961).



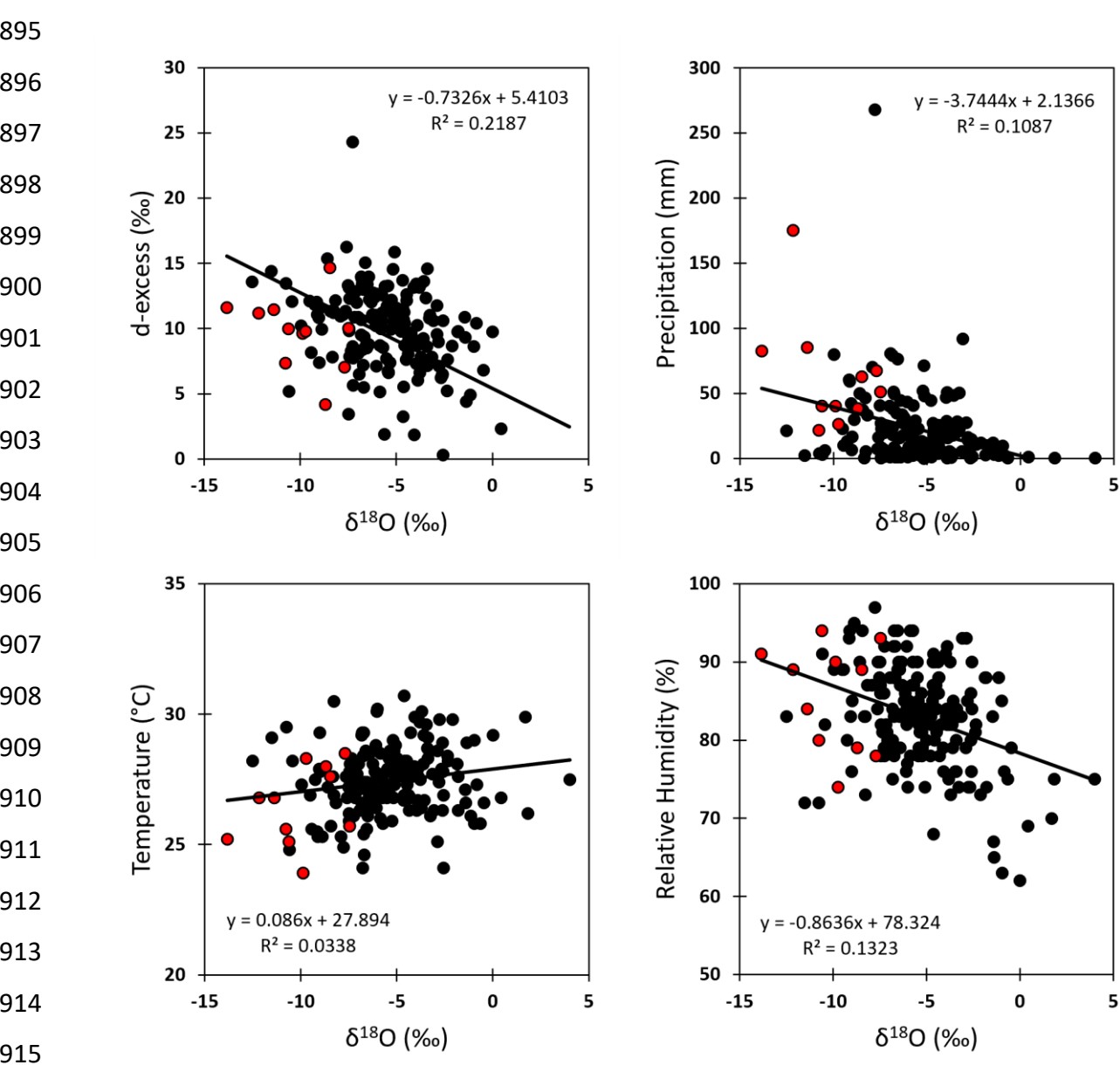





















**Figure 4 Correlations between daily δ¹⁸O values and daily values of d-excess, precipitation amount, temperature and
     relative humidity.** Linear regression line, correlation coefficient (R²), slope and intercept are shown in each plot. Samples
associated to TC are shown in red color similar to Figure 5.








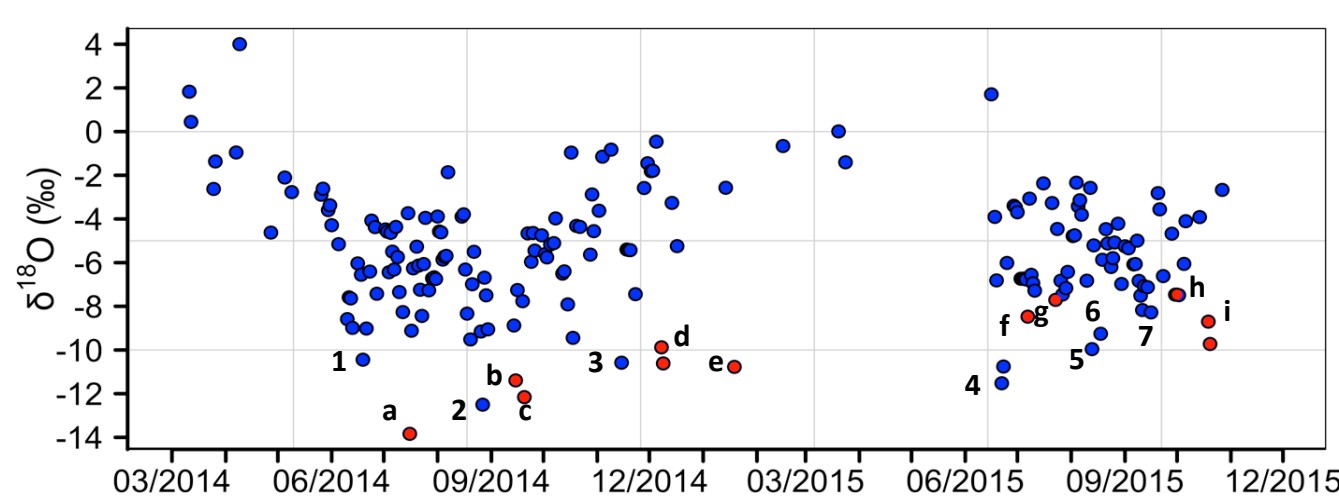

**Figure 5 Complete time series of 186 precipitation samples** taken between 10 March 2014 to 27 October 2015. δ $^{18}$O data points associated with TC activity are colored in red. Other anomalously low δ$^{18}$O values were investigated using IMERG satellite precipitation data. a: Rammasun 16/07/14, -13.84 ‰, 83 mm. b: Kalmaegi 15/09/14, -11.39 ‰, 85 mm. c: Fung-Wong 20/09/14, -12.16 ‰, 175 mm. d: Hagupit 8-9/12/14, -9.88 ‰, -10.62 ‰, 40 mm. e: Mekkhala 19/1/15, -10.77 ‰, 22 mm. f: Linfa 07/07/15, -8.5 ‰, 63 mm. g: Twelve 23/07/15, -7.7 ‰, 68 mm. h: Mujigae 01/10/15, -7.5 ‰, 51 mm. i: Koppu 19-20/10/15. -8.7 ‰, -9.72 ‰, 38 mm, 26 mm. 1: storm passing by 19/06/14, -10.44 ‰, 6 mm. 2: large rain areas 27/08/14, -12.5 ‰, 21 mm 3: storm passing by 15/11/14 -10.58 ‰, 3 mm. 4: large rain areas, 22-23/06/15 -10.76 ‰, -11.52 ‰, 2 mm, 4 mm. 5: heavy rainfall 13/08/15, -9.96 ‰, 80 mm. 6: heavy rainfall 18/08/15, -9.26 ‰, 13 mm. 7: local convection 16/09/15, -8.28 ‰, 47 mm.

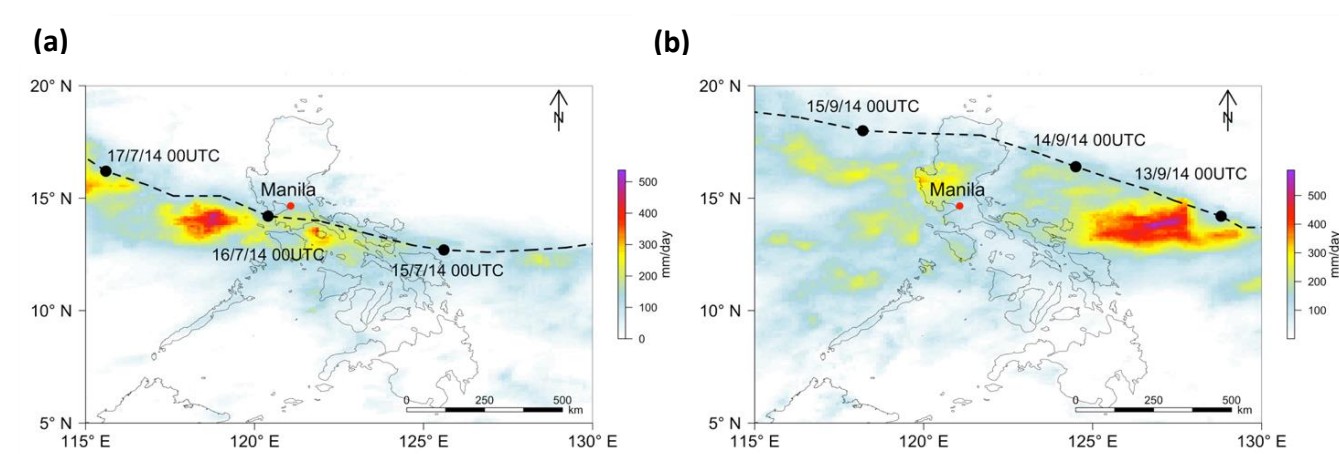

**Figure 6 Accumulated precipitation from IMERG satellite data and TC tracks from IBTrACKS** for a) Rammasun with precipitation accumulation for 14-17 July 2014, b) Kalmaegi with accumulated precipitation for 12-15 September 2014. Made with base layers from Natural Earth.

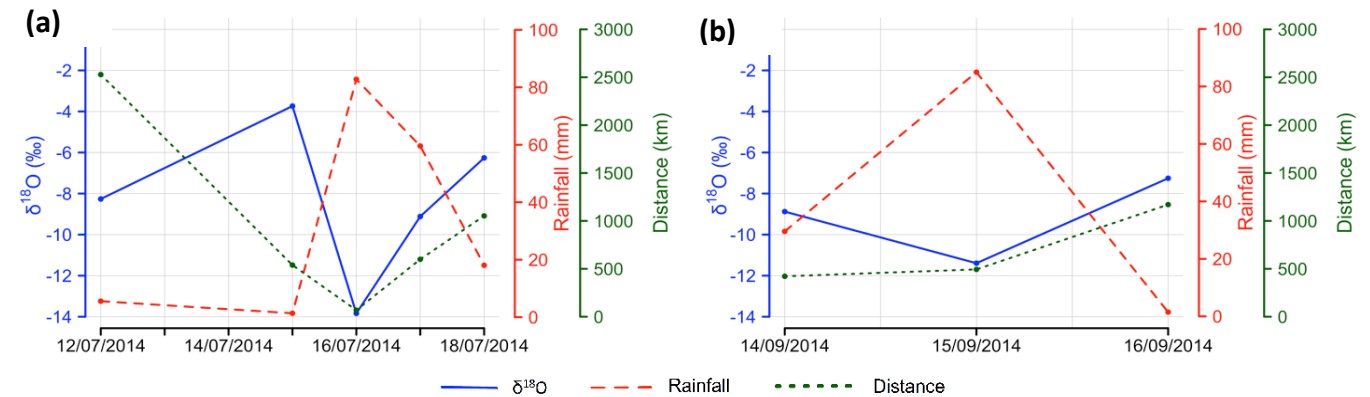

**Figure 7 Isotopic signature from TCs during their passage to the Metropolitan Manila sampling site.** $\delta^{18}O$ (blue color), distance from storm center to sampling location (green) and daily rainfall amount (red). a) Rammasun, b) Kalmaegi

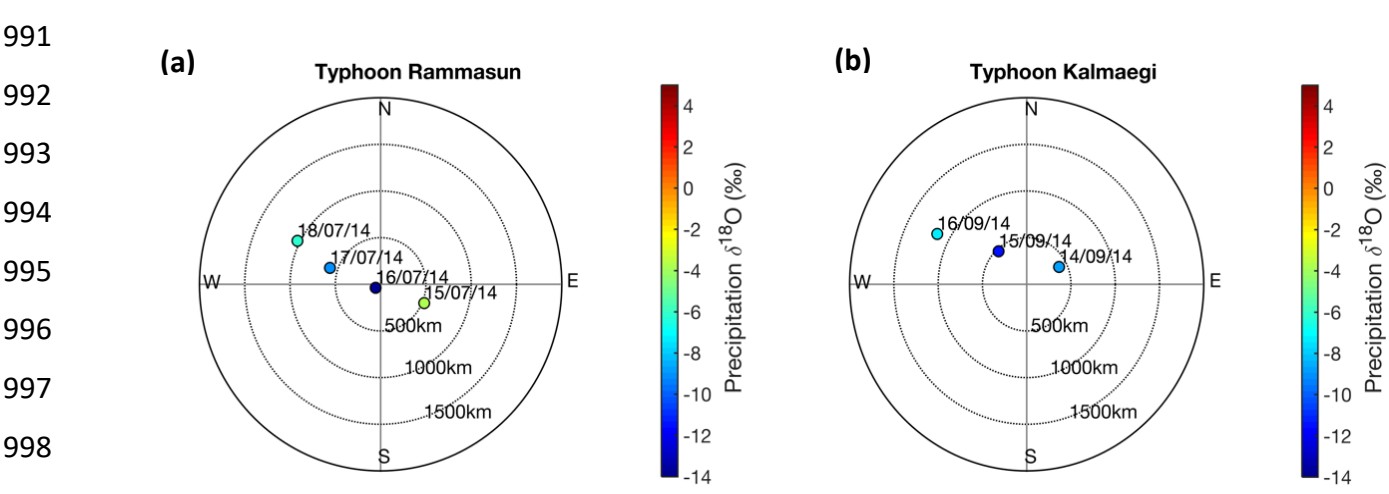

**Figure 8 Spatio-temporal evolution of δ¹⁸O isotopes**. Centered on Metropolitan Manila collection site, different radii provide information on distance between storm's center to Metropolitan Manila. δ¹⁸O values are color coded. a) Rammasun, b) Kalmaegi

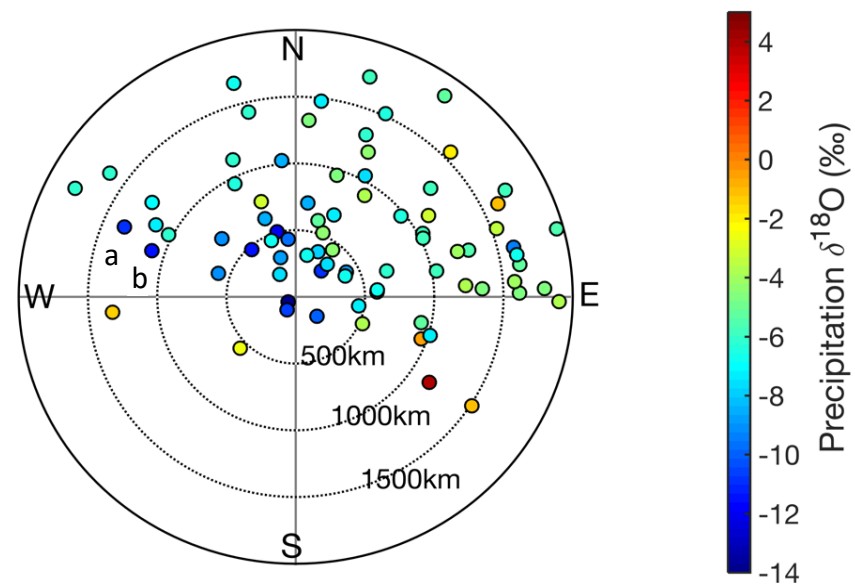

**Figure 9 Spatio-temporal variation of isotopes** related to TC activity within 2000 km, with different radii indicating the distance towards Metropolitan Manila. δ¹⁸O values are color coded.

**Tables**

**Table 1 Costliest typhoons in the Philippines.** Two devastating typhoons, Rammasun and Koppu (ranking 3 and 7), occurred during our study period and made landfall. Damage in USD based on each time of TC occurrence (not adjusted to current inflation rates).

| Rank | Name (local name) | Category (Saffir Simpson scale) | Period of occurrence | Damage in USD | Fatalities | Part of our dataset |
|---|---|---|---|---|---|---|
| 1. | Haiyan (Yolanda) | Category 5 | 2-11 November 2013 | ~ 2.06 billion USD | ~ 6000 | No |
| 2. | Bopha (Pablo) | Category 5 | 2-10 December 2012 | ~ 977 million USD | 1067 | No |
| 3. | Rammasun (Glenda) | Category 5 | 12-17 July 2014 | ~ 880 million USD | 106 | Yes |
| 7. | Koppu (Lando) | Category 4 | 12-21 October 2015 | ~ 310 million USD | 62 | Yes |
| References: Alojado and Padua, 2015; Lagmay et al., 2015; NDRRMC, 2012, 2014, 2015; Soria et al., 2016 | | | | | | |

**Table 2 Monthly average values of the 19-month time series** of $\delta^{18}$O, $\delta^2$H, d-excess and meteorological parameters (temperature and relative humdity) except precipitation values are reported as monthly totals.

| Month | $\delta^{18}$O (‰) | $\delta^2$H (‰) | d-excess (‰) | Precipitation (mm) | Temperature (°C) | Relative humidity (%) |
|---|---|---|---|---|---|---|
| Mar 14 | -0.43 | -0.62 | 2.82 | 19.2 | 27.1 | 70.0 |
| Apr 14 | -0.53 | -4.54 | -0.33 | 22.6 | 28.8 | 68.9 |
| May 14 | -2.89 | -13.50 | 9.63 | 99.1 | 29.8 | 71.7 |
| Jun 14 | -6.90 | -44.90 | 10.28 | 239.1 | 28.7 | 81.2 |
| Jul 14 | -6.46 | -41.68 | 10.04 | 455.4 | 27.5 | 86.6 |
| Aug 14 | -6.39 | -42.63 | 8.51 | 420.7 | 27.4 | 85.7 |
| Sep 14 | -7.29 | -48.57 | 9.76 | 654.9 | 27.4 | 85.3 |
| Oct 14 | -5.24 | -31.73 | 10.19 | 406.4 | 26.9 | 84.2 |
| Nov 14 | -4.39 | -27.64 | 7.48 | 90.5 | 26.9 | 80.0 |
| Dec 14 | -4.72 | -28.00 | 9.79 | 154.6 | 26.0 | 81.4 |
| Jan 15 | -6.67 | -44.41 | 8.97 | 29.2 | 24.6 | 77.8 |
| Feb 15 | -0.66 | -19.70 | -14.41 | 2.7 | 25.5 | 70.7 |
| Mar 15 | -0.70 | 3.95 | 9.54 | 6.6 | 26.8 | 62.9 |
| Apr 15 | | | | 64.8 | 29.1 | 62.0 |
| May 15 | | | | 74.6 | 29.7 | 68.4 |
| Jun 15 | -5.52 | -34.47 | 9.71 | 328.7 | 29.3 | 73.1 |
| Jul 15 | -6.04 | -36.69 | 11.61 | 28.6 | 27.8 | 80.5 |
| Aug 15 | -5.25 | -32.28 | 9.74 | 459.3 | 28.0 | 81.1 |
| Sep 15 | -6.12 | -37.07 | 11.86 | 444.8 | 28.0 | 81.0 |
| Oct 15 | -6.27 | -40.80 | 6.60 | 250.5 | 27.8 | 78.0 |