# Peer review of "1. Introduction"

_Natural Hazards and Earth System Sciences, 2019_

## Referee Comment (RC1) · Anonymous Referee #1 · 18 Mar 2020

In the manuscript, the authors attempted to discuss how a 19-month precipitation isotope dataset could be used by government agencies for mitigation and adaptation polices related to typhoons. The authors also suggested that this study could have possible implications for paleoclimate studies. The study is very ambitious. However, the 19-month dataset is difficult to provide substantial contributions to the hazards related to (1) precipitation processes during typhoon, (2) spatiotemporal isotope characteristics in the region, and (3) paleoclimate studies. The main problem of the paper is lack of scientific significance. By providing an extensive literature review in the introduction,

the authors tried to suggest that their 19-month isotope datasets could be useful for

(a) studying typhoon mechanisms

(b) showing spatiotemporal precipitation isotope signatures

(c) providing baseline data for paleoclimatology in the region

However, the above topics involve diverse spatial and temporal scales. It is very difficult to justify that the 19-month observations can inform all of them. In page 7, the authors provide excellent descriptions which summarise what the data could really tell us. The data can only tell us that the precipitation isotope values during typhoons were depleted. However, it is not really something very new. We did not learn anything about typhoon mechanisms apart form that the typhoon rainfalls were clearly depleted of the heavier isotopes. Turning to learning new spatiotemporal signatures of the regions, the daily data set is just not able to capture the typhoon dynamics. In Figure 4, we can see clear that we cannot expect that 3-5 data points for a typhoon event can tell us much about typhoon dynamics. Overall, we did not see many substantial results related to typhoon processes here, because of the limitation of the data. Moreover, the work does not have strong materials related to hazard, although the authors tried to frame the work about hazard mitigation and adaptation policies (Page 2 Ln 40). This paper is more suitable to be a data paper instead of a research paper. Perhaps, the authors should think about submitting this paper to Earth System Science Data (ESSD)

---

## Author Comment (AC1) · 21 May 2020

**Precipitation stable isotopic signatures of tropical cyclones in Metropolitan Manila, Philippines show significant negative isotopic excursions**

Dominik Jackisch, Bi Xuan Yeo, Adam D. Switzer, Shaoneng He, Danica Linda M. Cantarero, Fernando P. Siringan, and Nathalie F. Goodkin

**Reply to reviewer's comments:**

Dear Reviewer, your comments are greatly appreciated. The following is our response regarding points that require clarification.

**In the manuscript, the authors attempted to discuss how a 19-month precipitation isotope dataset could be used by government agencies for mitigation and adaptation polices related to typhoons.**

Response: This is a misunderstanding of our point, where we proposed that (line 37-40) "There is a clear need for developing a better understanding of tropical cyclone (TC) dynamics and cyclone histories in the context of prediction that may allow government agencies to implement proper mitigation and adaptation policies."

We are not suggesting that our dataset is to be directly used by government agencies to base adaptation policies upon. Instead, we are highlighting the scientific significance of our study, that isotopic signals in precipitation can allow us to better understand typhoon dynamics, and inform paleoclimate studies and mitigation measures in the future. We will thus amend this sentence to bring across our point clearer.

**The authors also suggested that this study could have possible implications for paleoclimate studies. However, the 19-month dataset is difficult to provide substantial contributions to the hazards related to (1) precipitation processes during typhoon, (2) spatiotemporal isotope characteristics in the region, and (3) paleoclimate studies.**

Response: We affirm that our studies adds value in the understanding of tropical cyclone hazards in the Philippines, as our in-situ dataset captures information on precipitation isotope signals for normal precipitation events and tropical cyclone events – for an area such as the Philippines where the amount of studies on typhoon precipitation isotope signals are scant, despite the region being a hotspot for typhoon activities.

Being the first study investigating typhoon-related precipitation isotope signals in the Philippines, firstly we are able to learn about how precipitation isotope signals in this region respond towards typhoon events of different intensity while they pass through sampling site.

Second, we gain spatiotemporal information of captured typhoon events through studying the tropical cyclone's position and distance from the sampling station's location, and observing effects of variables such as typhoon distance and intensity on level of isotopic (e.g figure 4, 5, 6).

Third, our 19-month dataset provides modern-day baseline data for investigating typhoon activities in paleoarchives for the Philippines, or for paleoarchive data points near our sampling station at the very

least. As there is yet no precedent dataset or studies that can provide such baseline data for studying tropical cyclones in palaeoarchives for the Philippines based on precipitation isotopes, we are convinced that our study is able to make a substantial initial contribution to this area. Kindly refer to the following sections in our paper (lines 71-81, 388-414, 462-465) which discusses our findings' potential for paleoclimate applications.

**However, the above topics involve diverse spatial and temporal scales. It is very difficult to justify that the 19-month observations can inform all of them.**

Response: We recognize that the observations from our research would be unable to be applied to a spatial range as extensive as the whole of the Philippines. However, our findings are still able to inform and be utilized within a reasonable area in the Philippines, proximate to our sampling station. Also, with reference to our research aims in lines 103-107, the main contribution of our findings is that we are able to capture the influence of spatial distance and progression of tropical cyclones on precipitation isotopic signals at our sampling station. Although we may consider increased sampling sites and frequency for improved data resolution (spatially and temporally), our findings are nevertheless an initial contribution to such studies which are unprecedented for the Philippines, specifically Metropolitan Manila, and our research aims remain within the capability of our dataset.

**In page 7, the authors provide excellent descriptions which summarise what the data could really tell us. The data can only tell us that the precipitation isotope values during typhoons were depleted. However, it is not really something very new.**

Response: As outlined in our research aims (lines 103-107), not only is our data able to show that typhoon events result in depleted isotope signals – we were able to study how distance from the typhoon and storm intensity influenced the degree to which the precipitation isotope signals were depleted, for a region where research on this matter is scarce. This is discussed in great detail in the points above.

In addition, our data provided interesting findings that may indicate potential for further research in the future (see conclusions line 416 onwards). For instance, we found that strong local convective activities can induce "false signals" where precipitation isotopes are slightly more depleted than average, despite typhoons being distant (lines 316-318, 363-366). Furthermore, our findings indicated that rain-out history and topography may have effects on isotopic signals (lines 290-292, 326-331). Thus, our study provide differentiated findings on the extent to which typhoon variables can affect precipitation isotope signals, and we further discussed potential influences specific to the Philippines region, such as topography, that could influence precipitation isotope signals.

**We did not learn anything about typhoon mechanisms apart form that the typhoon rainfalls were clearly depleted of the heavier isotopes. Turning to learning new spatiotemporal signatures of the regions, the daily data set is just not able to capture the typhoon dynamics. In Figure 4, we can see clear that we cannot expect that 3-5 data points for a typhoon event can tell us much about typhoon dynamics.**

Response: We reiterate that our research objectives, as stated above, did not include the intention to extensively study typhoon dynamics with our data. Instead, our daily precipitation isotope measurements are suitable for analyzing certain typhoon characteristics, such as typhoon strength, track and rainfall intensity, and how this can have implications on paleoclimate studies for the region, for instance. We have also discussed the limitations of our dataset, and provided suggestions for obtaining better sampling frequencies in further studies. As for Figures 4 and 5, it is intended to show the extent of precipitation isotopic response to typhoon events, depending on the typhoon's proximity and rainfall, which is in line with our research aim – to better understand how typhoon events are captured in precipitation isotopes in the Philippines.

**Overall, we did not see many substantial results related to typhoon processes here, because of the limitation of the data.**

Response: The aim of our study is clearly stated (lines 102-107) and investigated with our dataset. Further, we get a first understanding about typhoon activities with isotopes, as no study has been done there before.

**Moreover, the work does not have strong materials related to hazard, although the authors tried to frame the work about hazard mitigation and adaptation policies (Page 2 Ln 40).**

Response: We clarify that the intention of our paper is not to frame the work around hazard mitigation and adaptation policies. Rather, we posit that our findings can be used to improve interpretation of results for typhoon paleoclimatology, specifically in the Philippines. Further, we suggest more studies similar to ours should be carried out, to mend the data gap and provide a better picture of the paleohistorical record of typhoons in the Philippines. This in turn can better inform hazard mitigation and adaptation for typhoons in the Philippines.

**This paper is more suitable to be a data paper instead of a research paper. Perhaps, the authors should think about submitting this paper to Earth System Science Data (ESSD).**

Response: We appreciate the reviewer's kind suggestion regarding the nature of our research paper. Despite the fact that studying typhoon dynamics extensively was not an intended research aim of our paper, we are still confident that our findings can be a contribution of scientific significance, as it provides insight into the influence of typhoon activity on precipitation isotopes in an understudied region, using a unique dataset. In addition, we also discuss on how other factors such as local convective rainfall and topography may influence precipitation isotope signals. As such, we trust that our contribution is better justified as a research paper rather than a data paper.

---

## Referee Comment (RC2) · Anonymous Referee #2 · 1 Sep 2020

General Comments

In this paper, Jackisch et al. use shifts in $\delta$18O values over a period of 19 months to look for tropical cyclone (TCs) signals in precipitation. This length of records may not be enough for a robust baseline, but still shows some interesting results which can be useful to better understand tropical cyclones in Southeast Asia.

Although the use of isotopes to reconstruct TC signals is not new, I believe that research studies like this help reinforce and learn more about patterns and the use of O

isotopes in paleotempestology in different regions and at different latitudes. This study also strengthens the fact that we may still be a way from using O isotope depletion as a reliable (or individual) proxy for TCs.

I think there may be a slight disconnect between this study and the use of O isotopes in paleotempestology. The authors discuss paleo reconstructions using isotope depletion (eg. Miller et al., 2006; Frappier et al. 2007) but then conclude "Based on our findings we conclude that the location of sample collection needs to be chosen strategically." When reconstructing paleo storms, researchers may not know or have geological evidence of precise movement and path of a TC. Making it potentially difficult to differentiate TCs from other precipitation events. See Oliva et al. (2017) for use of these proxies in plaeotempestology.

Specific Comments

Line 43. Ensure this is still true, I believe it is widely accepted that there is likeliness in increase in intensity but not necessarily in frequency. See Woodruff et al. (2013) "At the end of the twenty-first century there will probably be fewer, but stronger, storms globally." Also see IPCC.

Line 52-56 Same as above. Also a graph or figure could be helpful to visualize this.

Section 2.1. This section does not describe the sampling sites, it describes the Philippines. I am more interested about details of sample locations. Section 2.2 talks about sampling at $14.654°$ N, $121.068°$ E- Were there any obstructions? Any other potential sources of contamination? Was it on a roof or at ground level? Near other potential sources of water?

Line 205. Figure 2 shows that all nine typhoons left distinct, or at least depleted isotope signatures. Why are they not all in the results? The way it is written, it seems like Rammasun and Kalmaegi, along maybe with Hagupit are the only ones to leave such a signature. You hint at the reason at Line 336 but the values should still be presented

objectively in the results.

Line 226. What were the values? 'relatively isotopically enriched' does not mean much.

Line 301. I do not think you can consider these outliers, there are more of these values than ones associated to TCs.

Technical Corrections

General comment. Author should review and ensure the use the units and symbols. For example, the authors use d°m' at line 115 and dd.ddd° at line 132.

Line 35. A reference here would be helpful to support such a statement.

Line 40. "Nine TCs per year made landfall on [...] Philippine waters is 19.4 per year." Consider revising wording, slightly confusing.

Line 123. add year of census to population.

Line 134-137. I suggest removing commercial URLs. It is enough to say the Brand and model.

Line 149; 158 URL should be in reference list, not in-text.

Lines 343 -347. I believe you mean r2 (not r), also should all be in presented the same way, not some in-text and some in parentheses.

Section 4.4, and in general. The discipline of using paleoarchives to reconstruct TC activity is called paleotempestology.

———————————————

---

## Referee Comment (RC3) · Anonymous Referee #3 · 8 Oct 2020

In this manuscript, daily measurements of the isotopic composition of precipitation in Manila are presented that have been performed over a period of about 19 months. Events with strong isotopic depletion are linked to passages of tropical cyclones. Unfortunately, in my opinion, the paper is very limited in terms of scientific content (data analysis and interpretation). I have a hard time identifying novel results or conclusions that may merit publication in a peer-reviewed paper. I thus cannot recommend this study for publication in NHESS.

Specific comments:

[Figure]

- The only conclusion that really follows from the analyses presented in this manuscript is that the precipitation associated with tropical cyclones, in particular those passing relatively close to measurement site, is isotopically more depleted than precipitation from other cloud systems. However, this conclusion is not novel. Already in the late 1990ies, Lawrence and others obtained similar results based on more detailed measurements and analyses. I cannot think of any reason why this very general result should be particularly different or noteworthy for TCs in the Philippines. Moreover, this conclusion and the analyses in the manuscript correspond well with the isotopic amount effect (see next point), which has been widely discussed in the literature since Dansgaard's work in the 1960ies.

- A characteristic property of tropical precipitation is that larger precipitation amounts are associated with more depleted isotope ratios (amount effect). This is mentioned in passing in the introduction of this manuscript, but not discussed in detail. Nevertheless, it can explain the results presented here: As TCs typically lead to large precipitation amounts, it is to be expected that they are also associated with lower isotope ratios. This is hinted at in the manuscript, but not shown explicitly (e.g., by plotting precipitation amount against isotope ratio). Along the same line, precipitation amount typically declines with distance from the TC center (beyond the eyewall), as does isotopic depletion. As mentioned before, the fact that the results and interpretation do not go beyond this variant of the well-known amount effect strongly compromises the novelty of the study.

- A major motivation for the authors appears to come from potential applications of isotope data from proxy archives for paleoclimate reconstructions. However, I don't see how their data could add to the present practice of using tropical data for reconstructions of precipitation amount, based on the amount effect described above. There are many vague statements in the manuscript that, at least for me, are difficult to follow. For example, how could changes in TC intensity, frequency or distance from the proxy site be distinguished from single isotope time series? Why should an isotope time series only represent changes in TC precipitation and not, for instance, changes in the precipitation amount in non-TC time periods? If the idea should be to learn something about such more detailed atmospheric processes (related, e.g., to TCs) by combining proxy records from different locations, then this approach should also be demonstrated with the help of a contemporary study combining data distributed in space, and not just from a single location.

---

## Author Comment (AC2) · 12 Oct 2020

**Precipitation stable isotopic signatures of tropical cyclones in Metropolitan Manila, Philippines show significant negative isotopic excursions**

Dominik Jackisch, Bi Xuan Yeo, Adam D. Switzer, Shaoneng He, Danica Linda M. Cantarero, Fernando P. Siringan, and Nathalie F. Goodkin

**Reply to reviewer's comments:**

**General Comments:**

**In this paper, Jackisch et al. use shifts in δ18O values over a period of 19 months to look for tropical cyclone (TCs) signals in precipitation. This length of records may not be enough for a robust baseline, but still shows some interesting results which can be useful to better understand tropical cyclones in Southeast Asia. Although the use of isotopes to reconstruct TC signals is not new, I believe that research studies like this help reinforce and learn more about patterns and the use of O isotopes in paleotempestology in different regions and at different latitudes. This study also strengthens the fact that we may still be a way from using O isotope depletion as a reliable (or individual) proxy for TCs. I think there may be a slight disconnect between this study and the use of O isotopes in paleotempestology. The authors discuss paleo reconstructions using isotope depletion (eg. Miller et al., 2006; Frappier et al. 2007) but then conclude "Based on our findings we conclude that the location of sample collection needs to be chosen strategically." When reconstructing paleo storms, researchers may not know or have geological evidence of precise movement and path of a TC. Making it potentially difficult to differentiate TCs from other precipitation events. See Oliva et al. (2017) for use of these proxies in plaeotempestology.**

Response: Dear reviewer, thank you very much for your detailed and helpful review, which is very useful for improving our paper. We greatly appreciate your feedback and incorporated your comments and remarks into the manuscript.

Regarding your general comment, we thank you for highlighting the scientific significance and scientific contribution of our work. We agree with your remark that researchers may not know the precise path of a TC, but we do no think that our conclusion is a contradiction to this. However, such limitations are now included in the manuscript with reference to the work of Oliva et al. (2017) and we added the following at line 396: Nevertheless, it is important to consider possible limitations at the study site that arise in paleotempestology, such as sea level change or disruption of sedimentological records through floods or tsunamis. These need to be evaluated when comparing precipitation isotopes related to TCs with other proxy records such as speleothems and coastal deposits and when choosing the study area (Oliva et al., 2017).

At line 73 we added the reference of Oliva et al., 2017.

**Specific Comments:**

**Line 43. Ensure this is still true, I believe it is widely accepted that there is likeliness in increase in intensity but not necessarily in frequency. See Woodruff et al. (2013)"At the end of the twenty-first century there will probably be fewer, but stronger, storms globally." Also see IPCC.**

**Line 52-56 Same as above. Also a graph or figure could be helpful to visualize this.**

Response: Thank you, we rewrote this accordingly and changed it to the following two statements at line 42 and 52: Changing climate and associated warming of the surface ocean, will likely increase the intensity of tropical cyclones in the future (Emanuel, 2005; Webster and Holland, 2005; Woodruff et al., 2013). Eighty percent of the strongest typhoons making landfall in the Philippines over the last three decades developed during higher than average sea surface temperatures (SST), which supports evidence that TC intensities are projected to rise in the future due to an increase in global temperatures (Guan et al., 2018; Webster and Holland, 2005; Takagi and Esteban, 2016).

**Section 2.1. This section does not describe the sampling sites, it describes the Philippines. I am more interested about details of sample locations. Section 2.2 talks about sampling at 14.654◦N, 121.068◦E- Were there any obstructions? Any other potential sources of contamination? Was it on a roof or at ground level? Near other potential sources of water?**

Response: Thank you for indicating this, we have added more specific information at line 133 for readers to get a better understanding of the sampling site: The rain collection station was installed on the rooftop of the Marine Science Institute (14°39'02.5"N, 121°04'08.6"E), which is centrally situated in the campus and surrounded by trees and various green spaces. The rooftop location proved ideal for rainwater collection as it allowed for unobstructed access to rainwater without any potential sources of contamination.

**Line 205. Figure 2 shows that all nine typhoons left distinct, or at least depleted isotope signatures. Why are they not all in the results? The way it is written, it seems like Rammasun and Kalmaegi, along maybe with Hagupit are the only ones to leave such a signature. You hint at the reason at Line 336 but the values should still be presented objectively in the results.**

Response: Thank you for noticing this. The values are already presented in the result section with reference to figure 2. However, we have made it clearer and now show the isotope value in the text for each TC at lines 196, 197, 201 and 202. We further added the following section at line 202: The other TCs that occurred during the study period and were investigated by us were Mekkhala (Fig.2, point e, -10.77 ‰), Twelve (Fig.2, point g, -7.7 ‰) and Mujigae (Fig. 2, point h, -7.5 ‰).

**Line 226. What were the values? 'relatively isotopically enriched' does not mean much.**

Response: Thank you, we have added them correspondingly at line 226: As the Rammasun storm center tracked towards the northwest and away from Metropolitan Manila, our precipitation samples were relatively isotopically enriched for the following two days, namely -9.12 ‰ on 17 July and -6.26 ‰ on 18 July.

**Line 301. I do not think you can consider these outliers, there are more of these values than ones associated to TCs.**

Response: Thank you, this is correct. These outliers are not considered as they are not related to TC activities. We had identified these outliers as produced by convective precipitation events using IMERG satellite data.

**Technical Corrections:**

**General comment. Author should review and ensure the use the units and symbols. For example, the authors use d∘m' at line 115 and dd.ddd∘at line 132.**

Response: Thank you, we have revised it throughout the manuscript and made it uniform to d∘m'.

**Line 35. A reference here would be helpful to support such a statement.**

Response: We agree and have added Cinco et al., 2014 as reference.

**Line 40. "Nine TCs per year made landfall on [...] Philippine waters is 19.4 per year." Consider revising wording, slightly confusing.**

Response: Thank you, we changed it to the following: Nine TCs per year made landfall on average between 1951 to 2013 in the Philippines. The number of TCs not making landfall but reaching Philippine waters is substantially higher with 19.4 per year (Cinco et al., 2016).

**Line 123. add year of census to population.**

Response: Thank you for the suggestion, we have added the year of census at line 123: 101 million 2017 census

**Line 134-137. I suggest removing commercial URLs. It is enough to say the Brand and model.**

Response: Thank you, we have removed commercial URLs at line 134 and 137.

**Line 149; 158 URL should be in reference list, not in-text.**

Response: Thank you, we have removed these URLs from the text at line 149 and 158.

**Lines 343 -347. I believe you mean r2 (not r), also should all be in presented the same way, not some in-text and some in parentheses. Section 4.4, and in general.**

Response: Thank you for indicating this, we have made it uniform and now present these values in text and not in parentheses at line 343, 345, 346 and 347.

**The discipline of using paleoarchives to reconstruct TC activity is called paleotempestology.**

Response: Thank you, we have put more emphasis on this and properly mention paleotempestology several times throughout the text. We have added it at line 79, 396, 402 and 463.

---

## Author Response (AR1)

**Precipitation stable isotopic signatures of tropical cyclones in Metropolitan Manila, Philippines show significant negative isotopic excursions**

Dominik Jackisch, Bi Xuan Yeo, Adam D. Switzer, Shaoneng He, Danica Linda M. Cantarero, Fernando P. Siringan, and Nathalie F. Goodkin

**Comments of reviewer 1 and authors' reply to every concern:**

*In the manuscript, the authors attempted to discuss how a 19-month precipitation isotope dataset could be used by government agencies for mitigation and adaptation polices related to typhoons.*

Response: This is a misunderstanding of our point, where we proposed that (line 37-40) "There is a clear need for developing a better understanding of tropical cyclone (TC) dynamics and cyclone histories in the context of prediction that may allow government agencies to implement proper mitigation and adaptation policies." We are not suggesting that our dataset is to be directly used by government agencies to base adaptation policies upon. Instead, we are highlighting the scientific significance of our study, that isotopic signals in precipitation can allow us to better understand typhoon dynamics, and inform paleoclimate studies and mitigation measures in the future. We will thus amend this sentence to bring across our point clearer.

*The authors also suggested that this study could have possible implications for paleoclimate studies. However, the 19-month dataset is difficult to provide substantial contributions to the hazards related to (1) precipitation processes during typhoon, (2) spatiotemporal isotope characteristics in the region, and (3) paleoclimate studies.*

Response: We affirm that our study contributes to our understanding of tropical cyclone hazards in the Philippines, as our in-situ dataset captures information on precipitation isotope signals for normal precipitation events and tropical cyclone events – for an area such as the Philippines, where the studies on typhoon precipitation isotope signals are scant, despite the region being a hotspot for typhoon activities. Being the first study investigating typhoon-related precipitation isotope signals in the Philippines, we are able to learn about how precipitation isotope signals in this region respond to typhoon events with different intensities. In addition, we gain spatiotemporal information of typhoon events through studying the tropical cyclone's position and distance from the sampling station's location, and observing effects of variables such as typhoon distance and intensity on level of precipitation isotopes (e.g figure 4, 5, 6). Furthermore, our 19-month dataset provides modern-day baseline data for investigating typhoon activities in paleoarchives for the Philippines, as well as for paleoarchive data points near our sampling station. As there is yet no precedent dataset or studies that can provide such baseline data for studying tropical cyclones in palaeoarchives

established for the Philippines based on water isotopes, we are convinced that our study is able to make a substantial initial contribution to this area. Kindly refer to the following sections in our paper (lines 71-81, 388-414, 462-465) which discusses our findings' potential for paleoclimate applications.

*However, the above topics involve diverse spatial and temporal scales. It is very difficult to justify that the 19-month observations can inform all of them.*

Response: We recognize that the observations from our research would be unable to be applied to a spatial range as extensive as the whole of the Philippines. However, our findings are still informative and can be used within a reasonable range in the Philippines, proximate to our sampling station. Also, with reference to our research aims in lines 103-107, the main contribution of our findings is that we are able to capture the influence of spatial distance and progression of tropical cyclones on precipitation isotopic signals at our sampling station. Although the significance of this study is lessened by our single sampling site with a limited data resolution (both spatially and temporally), our findings are nevertheless an initial contribution to such studies which are unprecedented for the Philippines, specifically Metropolitan Manila, and our research aims remain within the capability of our dataset.

*In page 7, the authors provide excellent descriptions which summarise what the data could really tell us. The data can only tell us that the precipitation isotope values during typhoons were depleted. However, it is not really something very new.*

Response: As outlined in our research aims (lines 103-107), not only is our data able to show that typhoon events result in depleted isotope signals – we were able to study how distance from the typhoon and storm intensity influenced the degree to which the precipitation isotope signals were depleted, for a region where related research is scarce. This is discussed in great detail in the points above. In addition, our data provided interesting findings that may indicate potential for further research in the future (see conclusions line 416 onwards). For instance, we found that strong local convective activities can induce "false signals" where precipitation isotopes are slightly more depleted than average, despite typhoons being distant (lines 316-318, 363-366). Furthermore, our findings indicated that rain-out history and topography may have effects on isotopic signals (lines 290-292, 326-331). Thus, our study provides differentiated findings on the extent to which typhoon variables can affect precipitation isotope signals, and we further discussed potential influences specific to the Philippines region, such as topography, that could influence precipitation isotope signals.

*We did not learn anything about typhoon mechanisms apart form that the typhoon rainfalls were clearly depleted of the heavier isotopes. Turning to learning new spatiotemporal signatures of the regions, the daily data set is just not able to capture the typhoon dynamics. In Figure 4, we can see clear that we cannot expect that 3-5 data points for a typhoon event can tell us much about typhoon dynamics.*

**Response**: We reiterate that we have no intention to extensively study typhoon dynamics with our data in this study. Instead, our daily precipitation isotope measurements are suitable for analyzing certain typhoon characteristics, such as typhoon strength, track and rainfall intensity, and how this can have implications for paleoclimate studies in this region. We have also discussed the limitations of our dataset, and provided suggestions for obtaining better sampling frequencies in further studies. As for Figures 4 and 5, it is intended to show the extent of precipitation isotopic response to typhoon events, depending on the typhoon's proximity and rainfall, which is in line with our research aim – to better understand how typhoon events are captured in precipitation isotopes in the Philippines.

*Overall, we did not see many substantial results related to typhoon processes here, because of the limitation of the data.*

**Response**: The aim of our study is clearly stated at lines 102-107 and investigated with our dataset ("to understand if there is an isotopic response of precipitation to TC activities in the Philippines, and if so – what signal do we measure and how is it represented spatially. Further, we aim to understand the isotopic variation with distance from the TC track"). Thus, we get a first understanding about typhoon activities with isotopes, as no study has been done there before.

*Moreover, the work does not have strong materials related to hazard, although the authors tried to frame the work about hazard mitigation and adaptation policies (Page 2 Ln 40).*

**Response**: We clarify that the intention of our paper is not to frame the work around hazard mitigation and adaptation policies. Rather, we posit that our findings can be used to improve interpretation of results for typhoon paleoclimatology, specifically in the Philippines. Such knowledge has important implications for the prediction of typhoon activities in the study area. Further, we suggest more studies similar to ours should be carried out, to mend the data gap and provide a better picture of the paleohistorical record of typhoons in the Philippines. This in turn can better inform hazard mitigation and adaptation for typhoons in the Philippines.

*This paper is more suitable to be a data paper instead of a research paper. Perhaps, the authors should think about submitting this paper to Earth System Science Data (ESSD).*

**Response**: We appreciate the reviewer's kind suggestion regarding the nature of our research paper. Despite the fact that studying typhoon dynamics extensively was not an aim of our paper, we are still confident that our findings can be a contribution of scientific significance, as it provides insight into the influence of typhoon activity on precipitation isotopes in an understudied region. In addition, we also discuss on how other factors such as local convective rainfall and topography may influence precipitation isotope signals. As such, we trust that our contribution is better justified as a research paper rather than a data paper.

**Comments of reviewer 2 and authors' reply to every concern:**

*In this paper, Jackisch et al. use shifts in δ$^{18}$O values over a period of 19 months to look for tropical cyclone (TCs) signals in precipitation. This length of records may not be enough for a robust baseline, but still shows some interesting results which can be useful to better understand tropical cyclones in Southeast Asia. Although the use of isotopes to reconstruct TC signals is not new, I believe that research studies like this help reinforce and learn more about patterns and the use of O isotopes in paleotempestology in different regions and at different latitudes. This study also strengthens the fact that we may still be a way from using O isotope depletion as a reliable (or individual) proxy for TCs. I think there may be a slight disconnect between this study and the use of O isotopes in paleotempestology. The authors discuss paleo reconstructions using isotope depletion (eg. Miller et al., 2006; Frappier et al. 2007) but then conclude "Based on our findings we conclude that the location of sample collection needs to be chosen strategically." When reconstructing paleo storms, researchers may not know or have geological evidence of precise movement and path of a TC. Making it potentially difficult to differentiate TCs from other precipitation events. See Oliva et al. (2017) for use of these proxies in plaeotempestology.*

**Response**: Thank you very much for your positive feedback.  Your comments are very helpful for improving the quality of our paper. Regarding your general comment, we thank you for highlighting the scientific significance and scientific contribution of our work. We agree with your remark that we may not know the precise path of a TC, but we do not think that our conclusion contradicts this. Such limitation has been included in the revised manuscript with reference to the work of Oliva et al. (2017) and we added the following at line 396: Nevertheless, it is important to consider possible limitations at the study site that arise in paleotempestology, such as sea level change or disruption of sedimentological records through floods or tsunamis. These need to be evaluated when comparing precipitation isotopes related to TCs with other proxy records such as speleothems and coastal deposits and when choosing the study area (Oliva et al., 2017). At line 73 we added the reference of Oliva et al.,2017.

**Specific Comments:**

*Line 43. Ensure this is still true, I believe it is widely accepted that there is likeliness in increase in intensity but not necessarily in frequency. See Woodruff et al. (2013)"At the end of the twenty-first century there will probably be fewer, but stronger, storms globally." Also see IPCC.*
*Line 52-56 Same as above. Also a graph or figure could be helpful to visualize this.*

**Response**: Thank you, we rephrased the sentence accordingly and changed it to the following two statements at line 42 and 52: Changing climate with associated warming of the surface ocean will likely increase the intensity of tropical cyclones in the future (Emanuel, 2005; Webster and Holland, 2005; Woodruff et al., 2013). Eighty percent of the strongest typhoons making landfall in the Philippines over the last three decades developed during higher than average sea surface temperatures (SST), which supports evidence that TC intensities are projected to rise in the future due to an increase in global temperatures (Guan et al., 2018; Webster and Holland, 2005; Takagi and Esteban, 2016).

***Section 2.1. This section does not describe the sampling sites, it describes the Philippines. I am more interested about details of sample locations. Section 2.2 talks about sampling at 14.654◦N, 121.068◦E- Were there any obstructions? Any other potential sources of contamination? Was it on a roof or at ground level? Near other potential sources of water?***

Response: Thank you for pointing this out. In the revised manuscript, we have added more specific information at line 133 for readers to get a better understanding of the sampling site: The rain collection station was installed on the rooftop of the Marine Science Institute (14°39'02.5"N, 121°04'08.6"E), which is centrally situated in the campus and surrounded by trees and various green spaces. The rooftop location proved ideal for rainwater collection as it allows unobstructed access to rainwater without any potential sources of contamination.

***Line 205. Figure 2 shows that all nine typhoons left distinct, or at least depleted isotope signatures. Why are they not all in the results? The way it is written, it seems like Rammasun and Kalmaegi, along maybe with Hagupit are the only ones to leave such a signature. You hint at the reason at Line 336 but the values should still be presented objectively in the results.***

Response: The values are already presented in the result section with reference to figure 2. However, to make it clearer, we have now shown the isotope value in the text for each TC at lines 196, 197, 201 and 202. We have further added the following section at line 202: The other TCs that occurred during the study period and were investigated by us were Mekkhala (Fig.2, point e, -10.77 ‰), Twelve (Fig.2, point g, -7.7 ‰) and Mujigae (Fig. 2, point h, -7.5 ‰).

***Line 226. What were the values? 'relatively isotopically enriched' does not mean much.***

Response: We have added the values correspondingly at line 226: As the Rammasun storm center tracked towards the northwest and away from Metropolitan Manila, our precipitation samples were relatively isotopically enriched for the following two days, namely -9.12 ‰ on 17 July and -6.26 ‰ on 18 July.

***Line 301. I do not think you can consider these outliers, there are more of these values than ones associated to TCs.***

Response: This is correct. These outliers are not considered as they are not related to TC activities. We had identified these outliers as produced by convective precipitation events using IMERG satellite data.

**Technical Corrections:**

***General comment. Author should review and ensure the use the units and symbols. For example, the authors use d◦m' at line 115 and dd.ddd◦ at line 132.***

Response: Thank you, and we have revised the manuscript and used the d◦m' consistently.

*Line 35. A reference here would be helpful to support such a statement.*

**Response**: We have added Cinco et al., 2014 as reference there.

*Line 40. "Nine TCs per year made landfall on [...] Philippine waters is 19.4 per year." Consider revising wording, slightly confusing.*

**Response**: We rephrased this sentence as the following: Nine TCs per year made landfall on average between 1951 to 2013 in the Philippines. The number of TCs not making landfall but reaching Philippine waters is substantially higher with 19.4 per year (Cinco et al., 2016).

*Line 123. add year of census to population.*

**Response**: We have added the year of census at line 123: 101 million 2017 census

*Line 134-137. I suggest removing commercial URLs. It is enough to say the Brand and model.*

**Response**: We have removed commercial URLs at line 134 and 137.

*Line 149; 158 URL should be in reference list, not in-text.*

**Response**: We have removed these URLs from the text at line 149 and 158.

*Lines 343 -347. I believe you mean r2 (not r), also should all be in presented the same way, not some in-text and some in parentheses. Section 4.4, and in general.*

**Response**: We have made it uniform and now present these values in text and not in parentheses at line 343, 345, 346 and 347.

*The discipline of using paleoarchives to reconstruct TC activity is called paleotempestology.*

**Response**: Thank you, we have put more emphasis on this and properly mention paleotempestology several times throughout the text. We have added it at line 79, 396, 402 and 463.

**Comments of reviewer 3 and authors' reply to every concern:**

*In this manuscript, daily measurements of the isotopic composition of precipitation in Manila are presented that have been performed over a period of about 19 months. Events with strong isotopic depletion are linked to passages of tropical cyclones. Unfortunately, in my opinion, the paper is very limited in terms of scientific content (data analysis and interpretation). I have a hard time identifying novel results or conclusions that may merit publication in a peer-reviewed paper. I thus cannot recommend this study for publication in NHESS.*

*The only conclusion that really follows from the analyses presented in this manuscript is that the precipitation associated with tropical cyclones, in particular those passing relatively close to measurement site, is isotopically more depleted than precipitation from other cloud systems. However, this conclusion is not novel. Already in the late1990ies, Lawrence and others obtained similar results based on more detailed measurements and analyses. I cannot think of any reason why this very general result should be particularly different or noteworthy for TCs in the Philippines. Moreover, this conclusion and the analyses in the manuscript correspond well with the isotopic amount effect (see next point), which has been widely discussed in the literature since Dansgaard's work in the 1960ies.*

*A characteristic property of tropical precipitation is that larger precipitation amounts are associated with more depleted isotope ratios (amount effect). This is mentioned in passing in the introduction of this manuscript, but not discussed in detail. Nevertheless, it can explain the results presented here: As TCs typically lead to large precipitation amounts, it is to be expected that they are also associated with lower isotope ratios. This is hinted at in the manuscript, but not shown explicitly (e.g., by plotting precipitation amount against isotope ratio). Along the same line, precipitation amount typically declines with distance from the TC center (beyond the eyewall), as does isotopic depletion. As mentioned before, the fact that the results and interpretation do not go beyond this variant of the well-known amount effect strongly compromises the novelty of the study.*

*A major motivation for the authors appears to come from potential applications of isotope data from proxy archives for paleoclimate reconstructions. However, I don't see how their data could add to the present practice of using tropical data for reconstructions of precipitation amount, based on the amount effect described above. There are many vague statements in the manuscript that, at least for me, are difficult to follow. For example, how could changes in TC intensity, frequency or distance from the proxy site be distinguished from single isotope time series? Why should an isotope time series only represent changes in TC precipitation and not, for instance, changes in the precipitation amount in non-TC time periods? If the idea should be to learn something about such more detailed atmospheric processes (related, e.g., to TCs) by combining proxy records from different locations, then this approach should also be demonstrated with the help of a contemporary study combining data distributed in space, and not just from a single location.*

**Response**: Dear reviewer, we appreciate your feedback regarding our manuscript. We would like to maintain our position that our study provides novel results in the understanding of

typhoons and its effect on precipitation's isotopic composition in the Southeast Asian region. It is true that isotopic depletion related to tropical cyclones has been shown previously in other parts of the world. However, our study is the first of this kind in Southeast Asia, where there is still very limited data available, and provides insight into the magnitude of influence of typhoon events can have on precipitation isotopes.

In your comments, you mentioned the amount effect and how it might explain our data. However, our focus is not on the amount effect, and also we did not intend to investigate it further using our daily isotope measurements affected by typhoon activities. This is because the amount effect is not observed in daily precipitation isotope measurements, but rather on longer timescales such as months and years (please refer to Belgaman et al., 2016; He at al., 2018; Kurita et al., 2009; Marryanna et al., 2017; Permana et al., 2016). We introduced the amount effect (line 85) and provided the explanations for the isotopic depletion related to typhoons observed at our study site (lines 83 to 87). Yet, we believe that our daily precipitation isotope data is helpful for better understanding certain tropical cyclone characteristics in the Philippines, such as its rainfall intensity, strength, distance and track, and this may have implications for future studies in paleotempestology in the region.

To base analyses for other parts of the world such as Southeast Asia using the data from North/South America may result in inaccurate conclusions, as response of precipitation isotopes to typhoons likely varies from region to region because of different climate conditions. Our manuscript presents in-situ data of precipitation isotopes affected by TC activities in the Philippines, providing an important baseline for other studies such as paleotempestological investigations in Southeast Asia. Thus, we believe that our paper would add value to the study of paleotempestology in the Southeast Asian region, which can also help to draw conclusions for prospective mitigation measures in the long run. The results and discussion parts of the manuscript show the capabilities of an isotope time series of 19 months. For instance, we explain how distance (figure 4, figure 5) or TC frequency (figure 2) can be derived from an isotope time series. In addition, further analysis in our manuscript provides insights and caveats in the usage of precipitation isotopes for studies - such as potential "false alarms" for depleted isotope values, resulting from precipitation from other rainfall events rather than TC precipitation (lines 318, 365, 413, 458), as well as opportunity to improve future data quality for such studies by setting up several new stations covering a spatial gradient (line 432). The significance of our study is acknowledged by the second reviewer and the suggested changes have been made by us in order to improve the manuscript (for example presenting all the isotope values in the results or adding more information about the study site in Metropolitan Manila).

---

## Referee Report (RR1)

I believe the authors of the paper Precipitation stable isotopic signatures of tropical cyclones in Metropolitan Manila, Philippines show significant negative isotopic excursions, have adequately answered reviewer comments from the first round of reviews that are available in the public discussion forum. Barring a few editorial mistakes that should be fixed (ex. Line 135: Proved, vs. proofed), I believe this paper to be of scientific significance and brings to the community of paleotempestologists new evidence and information that will help support future work.

---

## Author Response (AR2)

**Precipitation stable isotopic signatures of tropical cyclones in Metropolitan Manila, Philippines show significant negative isotopic excursions**

Dominik Jackisch, Bi Xuan Yeo, Adam D. Switzer, Shaoneng He, Danica Linda M. Cantarero, Fernando P. Siringan, and Nathalie F. Goodkin

**Response to reviewer**

**General comment**

In this manuscript the authors analysed the isotopic signature of precipitation from March 2014 to October 2015 in Metropolitan Manila, Philippines. They found that rain water collected during the landfall of some tropical cyclones had a progressive more depleted isotopic composition compared to rain water sampled during other rainfall events. Based on the data analyses, the authors found a significant relation between the daily isotopic composition of precipitation and the distance of the tropical cyclones from the sampling site.

I think the authors presented an interesting and valuable dataset of the isotopic composition of precipitation in Metropolitan Manila, Philippines. Despite the interesting presentation of the data collected during the landfall of some tropical cyclones, there is still a lack of a thorough analysis of the dataset, and particularly of the isotopic fractionation, which was speculated in section 4, and the meteorological characteristics (e.g., rainfall amount, air temperature etc.) that could influence the isotopic signature of precipitation during the landfall of tropical cyclones and the other rainfall events (please see the specific comments). There is a vast literature on these topics, and the authors could give a read to manuscripts which presented analyses of long time series of precipitation isotopic data (e.g., Guo et al., 2021). Furthermore, I think the authors should have accepted the valuable indications of the previous reviewer 3 because most of those comments were feasible to implement in the revised manuscript.

In addition, I think the current version of the manuscript still lacks some clarity. For example, the authors should clearly present and organize the research questions and the specific objectives. At the moment, the results are presented in only one section, but the authors could split them into 2-3 sections, and move the results of the statistical relations from the discussion to the proper section of the results.

Response: Dear reviewer, we thank you very much for your helpful review and valuable suggestions. We appreciate your feedback and incorporated your remarks into the manuscript, which is now stronger as a result. The focus of our manuscript is not to study the isotopic variability throughout the seasons and to investigate the associated drivers of isotopic changes. Instead, the manuscript's aim is to get a better understanding of tropical cyclones' signals in precipitation in the Philippines and to provide much needed data for paleotempestology. However, we admit that a detailed presentation of our timeseries along with meteorological parameters provide additional information and adds value to our manuscript. We therefore added several new figures and provide more insights into the isotopic variability throughout the measurement period.

**Specific comments**

**- L104-105: Please clearly formulate the research questions.**

Response: We have rephrased these lines to clearly point out the objectives and research questions and organized them into bullet points at lines 103-109: The major objective of this research is the following: - To understand if there is an isotopic variation in precipitation associated to the TC landfall in the Philippines and if tropical cyclones leave clear isotopic signals. - To identify the isotopic signals measured for Metropolitan Manila and the intensity of the isotopic depletion associated to TC activities, and to identify how it is represented spatially. - To understand the isotopic variation with distance from the TC track in the Philippines.

- L105-106: Please clearly report the specific objectives of this research study. Response: Please see the response above.

- Section 3: Why is there only one sub-section? The authors could organize the results based on the specific objectives and move here some of the results presented in the discussion (e.g., L348-353). In addition, I suggest to provide in this section the results for  $\delta$ 2H, and improve the section by the analysis of deuterium excess (Dansgaard, 1964) and the relations between the two isotopes, and between them and some meteorological characteristics (e.g., rainfall amount and air temperature).

Response: Thank you for this helpful suggestion. As explained above, we did not intend to analyze in our study in detail the seasonal/atmospheric controls on isotopes, however we now provide more information on the relationship between isotopes and other meteorological parameters in order to display our 15-month timeseries. Additionally, we included several new figures (Figure 2, 3 and 4) and a table (Table 2) which help to visualize our dataset. We therefore added two new sections (3.1 and 4.2) and present these results at lines 201-243.

**- L241-243: Please expand the description of the results for Figure 5b.**

Response: We now describe Figure 5b in more detail at lines 294-297: This is also seen in a spatial representation in Fig. 8b, visualizing the track of Kalmaegi and the respective  $\delta^{18}$ O values. Kalmaegi was first approaching the sampling site on 14 September and passed away on 15 and 16 September. The lowest  $\delta^{18}$ O was measured on 15 September and is indicated in the figure in dark blue colour.

**- L247: I suggest to not use the term 'proxy' because stable isotopes cannot be assumed as representative of tropical cyclones. Please rephrase the title.**

Response: Thank you for highlighting this, we have changed it accordingly to "tracer" at line 301.

**- L253-254: I have not found the hypothesis in the introduction. The authors should add it or clearly address it.**

Response: As mentioned above, we have rephrased the research questions and now clearly address the hypothesis with the following at line 105: "... and if tropical cyclones leave clear isotopic signals".

**- L255-256: 'isotopic fractionation' was not assessed in the data analyses. I think the authors could significantly improve this manuscript by adding an analysis of deuterium excess, and of the dual-isotope plot (please see also my previous comments).**

Response: Thank you for the suggestion, as mentioned above, we have added new sub-sections and several figures accordingly.

**- L262 and L266-273: Please refer directly to the amount effect.**

Response: Thank you for the suggestion. We have now presented the amount affect in section 3.1. which is observed at our site on monthly timescales.

**- L304-306: Please refer to a specific figure.**

Response: We added reference to Figure 6 at line 358.

**- L312-317: Rainfall amounts are not shown in Figure 2, and these sentences are not well supported by the results.**

Response: Rainfall amounts are now shown in Figure 2 in order to provide additional information.

- L319: Why should we use stable isotopes of precipitation to detect or predict the landfall of tropical cyclones? It does not make sense to use isotopes in precipitation when satellite and radar data are available, particularly if the isotopic samples are collected at the daily timescale.

Response: We agree as seen at line 319, but our statement is referring to possible applications in paleotempestology (lines 372-374).

**- L331-332: Rainfall amount data should be shown in the figures and not only on the supplementary material.**

Response: We have added rainfall data along with various other parameters at Figure 2, in order to display the dataset in detail.

**- L424-425: It is very unclear why the authors should suggest the sampling of stalagmites or trees, instead of precipitation. Why in the same location should we expect a different isotopic composition in precipitation and water used by vegetation (assuming that vegetation exploits mainly waters associated to tropical cyclones)?**

Response: In this part we are referring to potential applications in paleotempestology, as for instance stalagmites might provide information on past cyclone activity from centuries ago. This is also a reference to the introduction section, where various studies are presented that use stalagmites or tree rings to study past cyclones.

**- L435-436: Distance cannot be considered as a factor controlling/determining the isotopic signature of precipitation.**

Response: Our analysis clearly shows that distance plays an important role, therefore we changed it to "influencing" at line 510.

**- Figure 2: Please add the time series of the main meteorological characteristics, i.e. daily rainfall and mean daily temperature measured at a local weather station.**

Response: We have added these parameters to Figure 2 which displays the whole timeseries of data.

**- Figure 4: Please show rainfall amount as vertical bars.**

Response: Thank you for the suggestion. We tried implementing rainfall amount as vertical bars, but realized that vertical bars do not work for this kind of figure and would be very confusing to the reader. We specifically chose lines for  $\delta^{18}$ O, distance and rainfall amount as this representation clearly visualizes the various spikes in data, such as the drop in  $\delta^{18}$ O value and a spike in rainfall with a drop in distance.

**- Figure 5 and 6: I suggest to connect the dots associated to the same tropical cyclone (this should help to understand the trajectories/direction of the tropical cyclones).**

Response: We also thank you for this suggestion, similarly, we tried to add this for figure 6 but unfortunately, having lines connecting the dots results in a figure that is very hard to understand. Connecting lines in figure 5 are redundant as the few points are clearly labeled with the date and therefore clearly show the cyclone trajectory.

**Technical corrections**

**- L24-25: 'isotopic response to tropical cyclones' is unclear. I suggest to replace with 'isotopic variation related to tropical cyclones'.**

Response: Thank you, we have changed it accordingly at lines 24-25.

**- L104: I suggest to rephrase as follows: 'if there is an isotopic variation in precipitation associated to the TC landfall'.**

Response: Thank you, it is now changed according to the suggestion at line 104.

**- L340: 'causes'; I am not sure the authors could refer to a cause-effect relation.**

Response: We have replaced the word 'causes' with 'produces' at line 387.

**- L349: Please put sample size, p-value and confidence interval between parentheses.**

Response: We have added parentheses accordingly at lines 426-430.

**- L365-366: Please refer to Figure 6 and clearly identify the two outliers in the figure.**

Response: We included a reference to Figure 6 "(see points a and b in Fig. 9)" and also marked the two outliers in the figure (line 443).

**- L372: 'a "false non-TC signal" of very negative rainfall unassociated with TC activity' is quite difficult to understand. I suggest to rephrase.**

Response: Thank you, we slightly rephrased the sentence at line 449: "... inducing a "false non-TC signal" of very negative  $\delta^{18}$ O which is not related to TC activity"

---

## Author Response (AR3)

**Precipitation stable isotopic signatures of tropical cyclones in Metropolitan Manila, Philippines show significant negative isotopic excursions**

Dominik Jackisch, Bi Xuan Yeo, Adam D. Switzer, Shaoneng He, Danica Linda M. Cantarero, Fernando P. Siringan, and Nathalie F. Goodkin

**Reply to reviewer's comments:**

**General comment**

**I thank the authors for considering my comments. I think the manuscript has improved significantly compared to earlier versions. However, I still have some more specific comments that should be addressed before accepting the manuscript.**

Response: Dear reviewer, we thank you very much for your further comments, which clearly helped us to further improve the quality of our manuscript. We agree with your suggestions and have made corresponding changes, and in particular, shortened up the conclusion part.

**Specific comments**
**L refers to the lines in the revised manuscript with the highlighted changes.**

**- L208-209: Please move the sentence to Section 2.2.**

Response: We have moved the sentence to Section 2.2. Isotopic data (lines 151-152).

**- L210-212: Based on my experience, I would say that there is a high temporal variability in d-excess. Indeed, some samples seem to have a very evaporative signature (negative d-excess), whereas others have a d-excess even larger than 10.**

Response: We have therefore slightly altered the sentences to: "Although d-excess shows relatively high temporal variability, ranging from -15.18 ‰ to 24.31 ‰, it largely clusters in a small range between 5 ‰ to 15 ‰." (lines 209-211)

**- L232: I do not understand why the authors computed an average monthly rainfall. I think that the monthly rainfall amount (i.e., the sum) should be reported in Table 2, as well as in the following sentences (and in the data analysis).**

Response: We have changed it accordingly and now report the total monthly rainfall in Table 2 and in the analysis.

**- L239: Please report in the text what 'r' is. Does it indicate the Pearson correlation coefficient?**

Response: As suggested, "r" is now described at line 237 as the Pearson correlation coefficient.

**- L411-413: These results are not evident based only on Table 2. I suggest presenting a figure**

**similar to Figure 4, but using monthly data. Please consider using monthly rainfall amounts, and not an average.**

Response: The monthly values are presented in Table 2. However, the correlation between monthly values of $\delta^{18}$O and $\delta^2$H, d-excess, temperature, relative humidity and rainfall was already reported in the results section at lines 237-242. Therefore, we added a reference to the results section at line 432 with "(see section 3.1)". Monthly rainfall is now reported as totals and not as an average.

**- L425-430: These lines belong to the Results. Please move them to the proper section.**

Response: We have moved these lines to the results section at lines 244-250.

**- Conclusions: This section seems too long, and I do think the main conclusions were presented already between L506 and L522. My advice is to remove the remaining text (or to shorten it), especially all the sentences presenting again in great detail the results and part of the discussion.**

Response: We shortened up the conclusion part. The initial part remains (lines 509-525) according to your suggestion, while the middle part is removed. We only kept the last few sentences as they sum up the manuscript.

**- Figure 4: I suggest showing the samples associated to TCs with a different color.**

Response: Thank you for highlighting this. Figure 4 now shows the samples associated to TCs (similar to Fig. 5) in red color.

**Technical corrections**
**- L103: It should be 'The main objectives of this research are the following:'**

Response: Thank you, we changed it accordingly at line 103.

**- L202 and L253: It is unclear what the authors mean with 'rainfall intensive'.**

Response: We changed it at line 203 and at line 261 to "heavy rainfall".